# Variation-Bounded Losses for Learning with Noisy Labels

## Abstract

The presence of noisy labels poses a significant challenge for training accurate deep neural networks. Previous works have proposed various robust loss functions designed to address this issue, which, however, often suffer from several drawbacks, such as underfitting or insufficient noise-tolerance. Furthermore, there is currently no reliable metric to guide the design of more effective robust loss functions. In this paper, we introduce the *Variation Ratio* as a novel metric to measure the robustness of loss functions. Leveraging this metric, we propose a new family of robust loss functions, termed *Variation-Bounded Losses* (VBL), characterized by a bounded variation ratio. We investigate theoretical properties of variation-bounded losses and prove that a smaller variation ratio would lead to better robustness. Additionally, we show that the variation ratio provides a more relaxed condition than the commonly used symmetric condition for achieving noise-tolerant learning, making it a valuable tool for designing effective robust loss functions. We modify several commonly used loss functions to the variation-bounded form. These variation-bounded losses are characterized by their simplicity, effectiveness, and theoretical guarantees. Extensive experiments demonstrate the superiority of our method in mitigating various types of label noise.

## 1 Introduction

In recent years, deep neural networks (DNNs) have made significant advancements across various machine learning tasks (LeCun et al., 2015; Han et al., 2020). However, the quest for superior performance often requires a significant investment in large, high quality annotated datasets. Unfortunately, obtaining accurate annotations is often prohibitively expensive and challenging. Noisy labels, which stem from human errors, imperfect labeling processes, or cognitive biases, are prevalent in real-world datasets and can significantly degrade model performance in supervised learning (Wei et al., 2021). Moreover, the ability to generalize from noisy supervision plays a pivotal role in the alignment of large language models (Burns et al., 2023). To address this issue, the development of noise-tolerant learning has become crucial in the realm of weakly supervised learning, attracting increasing attention and research focus. Among various approaches, the design of robust loss functions stands out due to its simplicity and universality (Zhang & Sabuncu, 2018; Ma et al., 2020; Wei et al., 2021; Zhou et al., 2023; Ye et al., 2023).

Previous works (Van Rooyen et al., 2015; Ghosh et al., 2017) theoretically proved that symmetric loss functions are inherently tolerant to symmetric and asymmetric label noise under some moderate assumptions. However, the fitting ability of symmetric loss functions is constrained by the overly strict symmetric condition (Zhou et al., 2021). Symmetric losses such as Mean Absolute Error (MAE) (Ghosh et al., 2017) and Normalized Cross Entropy (NCE) (Ma et al., 2020) have proven challenging to optimize (Ghosh et al., 2017; Ma et al., 2020). Some methods, such as Active Passive Loss (APL) (Ma et al., 2020) and Active Negative Loss (ANL) (Ye et al., 2023), use two symmetric losses simultaneously to improve the fitting ability, but some studies point out that they still converge slowly compared to common loss functions (Englesson & Azizpour, 2021). Thus, it is worth considering whether symmetric conditions should also be leveraged to design more effective robust loss functions.

Since symmetric loss functions are challenging to optimize, a common approach is to interpolate between symmetric MAE and fast-converging CE. Examples of such hybrid loss functions include Generalized Cross Entropy (GCE) (Zhang & Sabuncu, 2018), Symmetric Cross Entropy (SCE) (Wang

et al., 2019), Taylor Cross Entropy (Taylor-CE) (Feng et al., 2021), and Jensen-Shannon Divergence Loss (JS) (Englesson & Azizpour, 2021). However, this approach to enhancing fitting ability comes at the expense of robustness due to the CE component. Consequently, they fail to achieve complete noise-tolerance and are likely to overfit a part of noisy labels.

Recently, Zhou et al. (2021; 2023) have introduced another class of robust loss functions, known as Asymmetric Loss Functions (ALFs). They theoretically demonstrate that asymmetric loss functions exhibit noise-tolerance to clean-label-dominant noise. However, the proposed asymmetric losses, such as the Asymmetric Unhinged Loss (AUL), are overly complicated with numerous hyperparameters and often suffer from underfitting . Although they proposed the asymmetric condition, they did not provide a straightforward method for its implementation. To date, no design guidelines have been established to help create simpler and more efficient asymmetric loss functions.

In this paper, we introduce a new metric for loss functions, called the *Variation Ratio*, which measures the robustness of loss functions. Then, we propose a new family of robust loss functions, namely *Variation-Bounded Losses* (VBL), whose variation ratio is bounded. We provide comprehensive theoretical analyses of variation-bounded losses and demonstrate that the variation ratio is critical for both symmetric (Ghosh et al., 2017) and asymmetric (Zhou et al., 2021) conditions. From the symmetric condition perspective, we prove that a smaller variation ratio can lead to a tighter excess risk bound for various types of label noise. From the asymmetric condition perspective, we prove that if the variation ratio is less than a certain constant with respect to the label distribution, the variation-bounded loss becomes asymmetric and, consequently, noise-tolerant. Our analyses reveal that restricting the variation ratio is a more relaxed condition than the symmetric condition. This suggests that the variation ratio can serve as a valuable tool for designing more effective robust loss functions. Furthermore, we modify several commonly used loss functions to the variation-bounded form for practical applications.

The main contributions of our work are highlighted as follows:

- We introduce a novel metric, namely the *variation ratio*, to measure the robustness of loss functions, and propose a new family of robust loss functions, termed *Variation-Bounded Losses* (VBL), which have a bounded variation ratio.

- We provide comprehensive theoretical analyses of variation-bounded losses, demonstrating that a small variation ratio is essential for achieving noise-tolerant learning and offers a more relaxed condition compared to the symmetric condition.

- The variation ratio can serve as a valuable tool for designing more effective robust loss functions. We develop a series of concise variation-bounded losses. Extensive experiment results underscore the superiority of our method.

## 2 PRELIMINARY

**Problem Formulation.** Considering a classification problem, we denote $\mathcal{X} \subset \mathbb{R}^d$ as the sample space and $\mathcal{Y} = [K] = \{1, 2, ..., K\}$ as the label space, where $K$ is the number of classes. In the supervised scenario, a labeled dataset $\mathcal{S} = \{(\mathbf{x}_n, y_n)\}_{n=1}^N$ is typically available for training classifiers, where $(\mathbf{x}_n, y_n)$ are i.i.d draws from an underlying distribution $\mathcal{D}$ over $\mathcal{X} \times \mathcal{Y}$. The classifier $f$ is a model with a softmax layer, mapping from the sample space to the probability simplex, when the prediction label is $\hat{y} = \arg\max_k f(\mathbf{x})_k$. Specifically, the prediction function $f : \mathcal{X} \to \mathcal{U}$ estimates the prediction probability $\mathbf{u}(\cdot|\mathbf{x})$, where $\mathcal{U} = \{\mathbf{u} \in [0, 1]^K : \mathbf{1}^\top \mathbf{u} = 1\}$. Moreover, let $L : \mathcal{U} \times \mathcal{U} \to \mathbb{R}$ represent the classification loss $L(\mathbf{u}, \mathbf{e}_y)$, where $\mathbf{e}_y$ is the one-hot vector with its $y$-th element set to 1. In this paper, we consider the loss functional $\mathcal{L}$, where $\forall L \in \mathcal{L}$, $L(\mathbf{u}, \mathbf{v}) = \sum_{k=1}^K \ell(u_k, v_k)$ with a basic loss function $\ell$, where $u_k$ is the k-th element of the vector $\mathbf{u}$. For the sake of brevity, we might abbreviate $L(\mathbf{u}, \mathbf{e}_k)$ as $L(\mathbf{u}, k)$ in the following.

**Label Noise Model.** In the scenario of learning with noisy labels, we have access to a noisy training set $\tilde{\mathcal{S}} = \{(\mathbf{x}_n, \tilde{y}_n)\}_{n=1}^N$ rather than a clean counterpart $\mathcal{S}$. For the sample $\mathbf{x}$, the noise corruption process is described as the flipping of the true label $y$ into the label $\tilde{y}$ with a conditional probability $\eta_{\mathbf{x},\tilde{y}} = p(\tilde{y}|\mathbf{x}, y)$, where $\eta_{\mathbf{x}} = \sum_{k \neq y} \eta_{\mathbf{x},k}$ denotes the noise rate for $\mathbf{x}$. As in previous work (Ghosh et al., 2017; Xia et al., 2020; Ye et al., 2023), we mainly focus on three types of label noise as follows:

— *Symmetric noise*: $\eta_{\mathbf{x},y} = 1 - \eta$ and $\eta_{\mathbf{x},k \neq y} = \frac{\eta}{K-1}$,

— *Asymmetric noise:* $\eta_{\mathbf{x},y} = 1 - \eta_y$ and $\sum_{k \neq y} \eta_{\mathbf{x},k} = \eta_y$,

— *Instance-Dependent noise:* $\eta_{\mathbf{x},y} = 1 - \eta_{\mathbf{x}}$ and $\sum_{k \neq y} \eta_{\mathbf{x},k} = \eta_{\mathbf{x}}$.

Herein, for symmetric noise, noise rate $\eta_{\mathbf{x}} = \eta$ is a constant for any instance; for asymmetric noise, $\eta_{\mathbf{x}} = \eta_y$ denotes the noise rate for the $y$-th class; for instance-dependent noise, $\eta_{\mathbf{x}}$ denotes the noise rate for the instance $\mathbf{x}$. For asymmetric and instance-dependent noise, $\eta_{\mathbf{x},i}$ is not necessarily equal to $\eta_{\mathbf{x},j}$ for $i \neq j$.

**Noise-Tolerant Learning.** In the case of clean labels, the expected risk (Bartlett et al., 2006) with respect to the loss function $L \in \mathcal{L}$ and the prediction function $f$ is defined as $\mathcal{R}_L(f) = \mathbb{E}_{(\mathbf{x},y) \sim \mathcal{D}}[L(f(\mathbf{x}), y)]$. The goal of supervised learning is to identify the expectation risk minimizer $f^* \in \arg \min_{f \in \mathcal{F}} \mathcal{R}_L(f)$. However, in the presence of noisy labels, we would explicitly focus on minimizing the noisy risk as follows:

$$\mathcal{R}_L^{\eta}(f) = \mathbb{E}_{\mathcal{D}} \left[ (1 - \eta_{\mathbf{x}})L(f(\mathbf{x}), y) + \sum_{k \neq y} \eta_{\mathbf{x},k} L(f(\mathbf{x}), k) \right], \tag{2.1}$$

where $\sum_{k \neq y} \eta_{\mathbf{x},k} L(f(\mathbf{x}), k)$ is the noisy part that usually poses challenges in training accurate DNNs. As discussed in (Ghosh et al., 2017), a loss function $L$ is defined to be *noise-tolerant* if the global minimizer $f_{\eta}^* \in \arg \min_f \mathcal{R}_L^{\eta}(f)$ also minimizes $\mathcal{R}_L(f)$, i.e., $f_{\eta}^* \in \arg \min_f \mathcal{R}_L(f)$.

# 3 VARIATION-BOUNDED LOSSES

In this section, we provide a comprehensive description to our variation ratio and variation-bounded losses. Additionally, we present thorough theoretical analysis of variation-bounded losses, demonstrating that our method achieves robust and efficient learning in a concise manner. Detailed proofs are included in the Appendix.

## 3.1 DEFINITIONS

First, we introduce the concept of active and passive terms in a loss function, referred to as the active passive loss in (Ma et al., 2020).

**Definition 1** (Active and Passive Terms of Loss Functions). *For a loss function $L \in \mathcal{L}$, i.e., $L(\mathbf{u}, \mathbf{v}) = \sum_{k=1}^{K} \ell(u_k, v_k)$, we can simply decompose it as $L(\mathbf{u}, \mathbf{e}_y) = \ell(u_y, 1) + \sum_{k \neq y} \ell(u_k, 0)$. For brevity, we denote $\ell(u, 1)$ as $\ell_{active}(u)$ and $\ell(u, 0)$ as $\ell_{passive}(u)$.*

To enhance understanding, we further elaborate the active term $\ell_{active}$ and the passive term $\ell_{passive}$ through examples. For CE loss, the active term is $-\log u_y$, while the passive term is 0; in this case, we say that the loss has no passive term. For Mean Square Error (MSE), the active term is $\frac{1}{K}(1 - u_y)^2$ and the passive term is $\frac{1}{K} u_k^2$, for $k \neq y$.

Next, we introduce the proposed variation ratio and variation-bounded losses.

**Definition 2** (Variation Ratio). *For a loss function $L(\mathbf{u}, y) = \ell_{active}(u_y) + \sum_{k \neq y} \ell_{passive}(u_k)$, if $\ell_{active}$ is monotone decreasing, we define the variation ratio $v(L)$ as*

$$v(L) = \frac{\max_u |\nabla \ell_{active}(u)|}{\min_u |\nabla \ell_{active}(u)|}, \tag{3.1}$$

*where $u \in (0, 1)$ and $\nabla \ell = \frac{\partial \ell(u)}{\partial u}$ is the gradient of $\ell$ w.r.t. $u$.*

**Definition 3** (Variation-Bounded Losses). *If the variation ratio $v(L) < \infty$, the loss function $L$ is variation-bounded. Conversely, if $v(L) = \infty$, the loss function $L$ is variation-unbounded.*

For example, the variation ratio $v(L)$ of MAE is 1, indicating that MAE is a variation-bounded loss. Another example of a variation-bounded loss is Exponential Loss (EL), $L_{\text{EL}} = e^{-u_y}$, whose variation ratio is $e$. In contrast, $v(L)$ values are $\infty$ for CE, GCE ($q < 1$) and SCE ($\alpha > 0$), indicating that they are variation-unbounded losses.

In the subsequent sections, we will explore the detailed properties of variation-bounded losses.

## 3.2 SYMMETRIC CONDITION

Previous works (Van Rooyen et al., 2015; Ghosh et al., 2017) theoretically proved that a loss function is noise-tolerant to symmetric and asymmetric label noise under some mild conditions if it is symmetric (Symmetric Condition):

$$\sum_{k=1}^{K} L(\mathbf{u}, k) = C \tag{3.2}$$

where $C$ is a constant and $k \in [K]$ is the label corresponding to each class.

Because the symmetric condition in Eq. 3.2 is overly strict, symmetric losses are challenging to optimize (Ghosh et al., 2017; Zhou et al., 2021). A common method to address this issue is to interpolate between the symmetric mean absolute error (MAE) and the fast-converging cross-entropy (CE) (Zhang & Sabuncu, 2018; Wang et al., 2019; Englesson & Azizpour, 2021). However, this approach to enhancing fitting ability comes at the expense of robustness. Meanwhile, some studies (Ghosh et al., 2017; Li et al., 2020) have indicated that bounded losses are more robust than unbounded losses, with MSE[1] as a typical example of a bounded loss. The bounded property of the loss, $C_L \le L(\mathbf{u}, k) \le C_U$, ensures that $|\sum_{k=1}^{K} L(\mathbf{u}, k) - \sum_{k=1}^{K} L(\mathbf{v}, k)| \le K(C_U - C_L)$, where $\mathbf{u}$ and $\mathbf{v}$ are arbitrary vectors in the domain. This property brings the bounded loss closer to meeting the symmetric condition and enhances its robustness compared to unbounded losses such as CE. We can further derive this bounded property, making the boundary depend only on the variation ratio.

**Lemma 1.** *Consider a monotone decreasing basic loss $\ell_{active}$, for a loss function $L(\mathbf{u}, y) = c \cdot \ell_{active}(u_y)$, we have*

$$\left| \sum_{k=1}^{K} L(\mathbf{u}, k) - \sum_{k=1}^{K} L(\mathbf{v}, k) \right| \le v(L) - 1. \tag{3.3}$$

*where $c = \frac{1}{\min_u |\nabla \ell_{active}(u)|}$ is a normalization constant.*

$c$ in Lemma 1 is used to normalize the minimum absolute value of the gradient to 1, so that different loss functions can be compared on the same scale. This Lemma shows that a smaller $v(L)$ results in better symmetry. Specifically, when $v(L)$ reaches its minimum value of 1, the loss is symmetric. And this loss is essentially a linear function, representing a scaled version of the MAE.

Based on Lemma 1, we derive excess risk bounds (Bartlett et al., 2006) under various types of label noise. First, we prove the situation of symmetric noise.

**Theorem 1** (Excess Risk Bound under Symmetric Noise). *In a multi-class classification problem, if the loss function $L \in \mathcal{L}$ satisfies $|\sum_{k=1}^{K} L(\mathbf{u}, k) - \sum_{k=1}^{K} L(\mathbf{v}, k)| \le v(L) - 1$, then for symmetric noise satisfying $\eta < 1 - \frac{1}{K}$, the excess risk bound for $f$ can be expressed as*

$$\mathcal{R}_L(f_\eta^*) - \mathcal{R}_L(f^*) \le c(v(L) - 1), \tag{3.4}$$

*where $c = \frac{\eta}{(1-\eta)K - 1}$ is a constant, $f_\eta^*$ and $f^*$ denote the global minimum of $\mathcal{R}_L^\eta(f)$ and $\mathcal{R}_L(f)$, respectively.*

Next, we address the more complex situations of asymmetric and instance-dependent noise.

**Theorem 2** (Excess Risk Bound under Asymmetric and Instance-Dependent Noise). *In a multi-class classification problem, if the loss function $L \in \mathcal{L}$ satisfies $|\sum_{k=1}^{K} L(\mathbf{u}, k) - \sum_{k=1}^{K} L(\mathbf{v}, k)| \le v(L) - 1$, then for label noise $1 - \eta_{\mathbf{x}} > \max_{k \ne y} \eta_{\mathbf{x},k}$, $\forall \mathbf{x}$, if $\mathcal{R}_L(f^*)$ is minimum, the excess risk bound for $f$ can be expressed as*

$$\mathcal{R}_L(f_\eta^*) - \mathcal{R}_L(f^*) \le (1 + \frac{c}{a})(v(L) - 1), \tag{3.5}$$

*where $c = \mathbb{E}_\mathcal{D}(1 - \eta_{\mathbf{x}})$ and $a = \min_{\mathbf{x},k}(1 - \eta_{\mathbf{x}} - \eta_{\mathbf{x},k})$ are constants, $f_\eta^*$ and $f^*$ denote the global minimum of $\mathcal{R}_L^\eta(f)$ and $\mathcal{R}_L(f)$, respectively. For asymmetric noise, $\eta_{\mathbf{x}} = \eta_y$, and for instance-dependent noise, $\eta_{\mathbf{x}} = \eta_{\mathbf{x}}$.*

Theorem 1 and 2 demonstrate that a smaller variation ratio $v(L)$ results in more robust to label noise. Additionally, the excess risk bound can be controlled by the $v(L)$. Accordingly, variation ratio $v(L)$ can be used to measure the robustness of loss functions.

---

[1]Some works use MSE loss without adding softmax layer (Han et al., 2022); in this paper, we consider MSE with softmax layer (Ghosh et al., 2017).

## 3.3 ASYMMETRIC CONDITION

Previous works (Zhou et al., 2021; 2023) proposed asymmetric loss functions that are noise-tolerant for clean-label-dominant noise, i.e., $1 - \eta_{\mathbf{x}} > \max_{k \neq y} \eta_{\mathbf{x},k}, \forall \mathbf{x}$. However, the proposed asymmetric loss functions are too complex with many hyperparameters, and easily produce the underfitting problem. In this subsection, we revisit the asymmetric condition through the variation ratio.

**Definition 4** (Asymmetric Condition). *On the given weights* $w_1, \ldots, w_k \geq 0$, *where* $\exists t \in [K]$, *s.t.,* $w_t > \max_{k \neq t} w_k$, *a loss function* $L(u, k)$ *is called asymmetric if* $L$ *satisfies*

$$\arg\min_{\mathbf{u}} \sum_{k=1}^{K} w_k L(\mathbf{u}, k) = \arg\min_{\mathbf{u}} L(\mathbf{u}, t), \tag{3.6}$$

*where we always have* $\arg\min_{\mathbf{u}} L(\mathbf{u}, t) = \mathbf{e}_t$.

Zhou et al. (2021) proved that asymmetric loss functions are noise-tolerant for clean-label-dominant noise, i.e., $1 - \eta_{\mathbf{x}} > \max_{k \neq y} \eta_{\mathbf{x},k}, \forall \mathbf{x}$. Here, we prove that when the variation ratio $v(L)$ is less than a specific constant related to the label distribution, the variation-bounded loss is asymmetric.

**Theorem 3.** *On the given weights* $w_1, \ldots, w_k \geq 0$, *where* $\exists t \in [K]$ *and* $w_t > \max_{i \neq t} w_i$, *a monotone decreasing loss function* $L(\mathbf{u}, k) = \ell_{active}(u_k)$ *is asymmetric if (1)* $\frac{\partial^2 \ell_{active}(u)}{\partial u^2} \leq 0$ *or (2)* $v(L) \leq \frac{w_t}{w_i}$ *for any* $i \neq t$.

Condition (1) in Theorem 3 is not favorable for optimizing the loss function. Specifically, for CE loss, we have $\frac{\partial^2 \ell_{active}(u)}{\partial u^2} > 0$, while for MAE, we have $\frac{\partial^2 \ell_{active}(u)}{\partial u^2} = 0$. If $\frac{\partial^2 \ell_{active}(u)}{\partial u^2} < 0$, the loss function would be even harder to optimize than MAE. Therefore, in practice, such loss functions are generally not considered. Instead, we primarily focus on condition (2) in Theorem 3.

For a variation-bounded loss described in Theorem 3, if it satisfies $v(L) \leq \frac{1 - \eta_{\mathbf{x}}}{\max_{k \neq y} \eta_{\mathbf{x},k}}$, i.e., condition (2) in Theorem 3 for learning with noisy labels context, then the loss function is asymmetric. For example, consider a 10 classes dataset with 0.8 symmetric noise. If $v(L) \leq \frac{1 - \eta_{\mathbf{x}}}{\max_{k \neq y} \eta_{\mathbf{x},k}} = \frac{0.2}{0.8/9} \approx$ 2.25, then the loss function is asymmetric and therefore noise-tolerant to clean-label-dominant noise. Notably, this constitutes a more relaxed condition compared to the symmetric condition, because it only requires that $v(L) \leq 2.25$, whereas symmetric MAE requires $v(L)$ to equal the minimum value of 1. Thus, variation-bounded losses have better fitting ability than symmetric losses, enabling them to achieve both robust and efficient learning simultaneously.

## 3.4 VARIATION-BOUNDED LOSSES

In this subsection, we concisely generalize several commonly used loss functions to the variation-bounded form. We use $\mathbf{u} = f(\mathbf{x})$ to denote the prediction probability after the softmax layer, and $u_y$ is the predicted probability for the label class.

**Variation Cross Entropy (VCE):**

$$L_{\text{VCE}} = -\log(u_y + a), \tag{3.7}$$

where $a \geq 0$ is a hyperparameter. VCE is an extension of the CE loss. If $a > 0$, the variation ratio $v(L_{\text{VCE}}) = \frac{1+a}{a}$. If $a = 0$, $v(L_{\text{VCE}}) = \infty$ and this is the CE loss.

**Variation Exponential Loss (VEL):**

$$L_{\text{VEL}} = a^{-u_y}, \tag{3.8}$$

where $a > 1$ is a hyperparameter. VEL is an extension of the Exponential Loss (EL). The variation ratio $v(L_{\text{VEL}}) = a$. If $a = e$, this is the Exponential Loss.

**Variation Mean Square Error (VMSE):**

$$L_{\text{VMSE}} = \frac{1}{K} \|a \cdot \mathbf{e}_y - \mathbf{u}\|_2^2 = \frac{1}{K} [(a - u_y)^2 + \sum_{k \neq y} u_k^2], \tag{3.9}$$

where $a \geq 1$ is a hyperparameter. VMSE is an extension of the MSE loss. If $a > 1$, the variation ratio $v(L_{\text{VMSE}}) = \frac{a}{a-1}$. If $a = 1$, $v(L_{\text{VMSE}}) = \infty$ and this is the MSE loss.

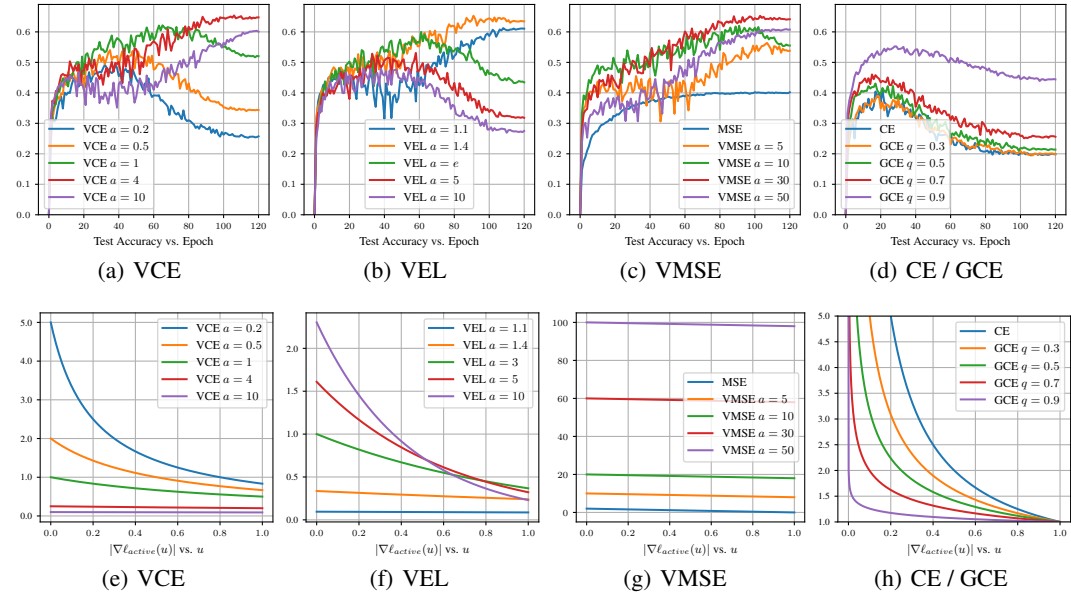

(a) VCE (b) VEL (c) VMSE (d) CE / GCE

(e) VCE (f) VEL (g) VMSE (h) CE / GCE

Figure 1: **Top (a-d):** Test accuracies on CIFAR-10 with 0.8 symmetric noise. **Bottom (e-h):** Absolute values of gradients, i.e., $|\nabla \ell_{active}|$. Among these loss functions, VCE, VEL and VMSE are variation-bounded; CE, GCE and MSE are variation-unbounded.

**Remark.** It is worth noting that Lemma 1 and Theorem 3 consider the loss function only have the active term, i.e., $L = \ell_{active}$. Hence, they are not applicable to VMSE, which includes both active and passive terms. Previous works (Zhou et al., 2021; 2023) also only considered asymmetric loss functions containing only the active term. To date, no studies have explored asymmetric loss functions that have both active and passive terms, because this is a more complex scenario. In this paper, we extend the asymmetric loss function to include both active and passive terms. We prove that by restricting the variation ratio $v(L_{\text{VMSE}})$, VMSE can satisfy the asymmetric condition.

**Corollary 1.** *On the given weights $w_1, \ldots, w_k$, where $w_m > w_n$, and $w_n = \max_{k \neq m} w_k$. The loss function $L_p(\mathbf{u}, y) = \frac{1}{K} \|a \cdot \mathbf{e}_y - \mathbf{u}\|_p^p = \frac{1}{K}[(a - u_y)^p + \sum_{k \neq y} u_k^p]$, where $p > 0$ and $a \geq 1$ are hyperparameters, is asymmetric if and only if $\frac{w_m}{w_n} \geq [v(L_p) + \frac{\sum_{k \neq m} \frac{w_k}{w_n}}{(a-1)^{p-1}}] \cdot \mathbb{I}(p > 1) + \mathbb{I}(p \leq 1)$.*

In this paper, for $p$ in Corollary 1, we use 2 for VMSE to be consistent with MSE. Corollary 1 demonstrates that VMSE, which has both active and passive terms, can satisfy asymmetric condition and subsequently become noise-tolerant.

**Association with Rescaled-MSE.** We observe that our VMSE is similar to the Rescaled-MSE (Hui & Belkin, 2021), i.e., $L = \frac{1}{K}[b \cdot (a - u_y)^2 + \sum_{k \neq y} u_k^2]$, where $a, b \geq 1$ are hyperparameters. If $b = 1$, VMSE is equal to Rescaled-MSE. Although their work does not focus on learning with noisy labels, Hui & Belkin (2021) found that increasing the value of $a$ improves the fitting ability of Rescaled-MSE, especially when the number of classes is large. According to Corollary 1, increasing the value of $a$ also improves the robustness, because a larger $a$ leads to a smaller variation ratio $v(L_{\text{VMSE}})$. These analyses show that our VMSE possesses desirable properties. By selecting a moderately large $a$, it can simultaneously enhance both fitting ability and robustness.

**More Comparison with Existing Asymmetric Loss Functions.**

— *Complexity:* Previous asymmetric loss functions (Zhou et al., 2021), AUL, AGCE, and AEL, are relatively complex as they are explicitly designed with reference to the asymmetric condition. In contrast, our variation ratio simplifies the asymmetric condition, resulting in loss functions that are simpler and more effective. Specifically, prior asymmetric loss functions all have two hyperparameters, whereas our VCE, VEL, and VMSE have only one hyperparameter.

— *Fitting Ability:* Since prior asymmetric losses and our VBL are theoretically proven to be noise-tolerant, the performance difference depends mainly on the fitting ability. Our VBL have a more

concise form, making them easier to adjust the hyperparameters and optimize. We'll show this clearly in the Experiments Section.

— *Expansibility:* Previous work (Zhou et al., 2021) can only be applied to loss functions containing only the active term, while our variation ratio can be applied to the loss function containing both active and passive terms, i.e., VMSE.

**Visualization and Hyperparameter Analysis.** Figure 1 presents test accuracies on CIFAR-10 with 0.8 symmetric noise and absolute values of gradients, $|\nabla \ell_{active}|$, for different loss functions. As can be seen, for variation-bounded losses (Figure 1(a), 1(b), and 1(c)), a smaller variation ratio ($a \uparrow$ for VCE and VMSE; $a \downarrow$ for VEL) can enhance robustness and achieve noise-tolerant learning. However, a too small variation ratio may reduce the fitting ability. Therefore, it is suggested to choose a moderate variation ratio to achieve both robust and efficient learning. Conversely, for variation-unbounded losses such as CE and GCE (Figure 1(d)), they cannot achieve complete noise-tolerance. Because for CE and GCE, $|\nabla \ell_{active}(u)| \to \infty$ as $u \to 0$ (Figure 1(h)), it follows that their variation ratio is $\infty$. As training progresses, CE and GCE overfit to label noise, resulting in a decrease in test accuracy.

### 3.5 Combination of NCE and VBL.

Recently, the most advanced robust loss functions often combine their proposed methods with Normalized Cross Entropy (NCE) (Ma et al., 2020). Notable examples include Active Passive Loss (APL) (Ma et al., 2020), Asymmetric Loss Functions (ALFs) (Zhou et al., 2021), and Active Negative Loss (ANL) (Ye et al., 2023). The combination of two different robust loss functions, can mutually enhance the optimization processes of each other, thus improving the overall fitting ability of the model. These works (Ma et al., 2020; Zhou et al., 2021; Ye et al., 2023) have experimentally demonstrated that combining NCE often results in improved performance.

Inspired by them, our proposed VBL can also be combined with NCE to obtain better performance. We formulate the combination of NCE and VBL as follows

$$L_{\text{NCE+VBL}} = \alpha \cdot L_{\text{NCE}} + \beta \cdot L_{\text{VBL}} \tag{3.10}$$

We can easily prove that NCE+VBL is still noise-tolerant. Previous work (Zhou et al., 2021) proved that symmetric loss functions are completely asymmetric, and the combination of asymmetric loss functions remains asymmetric. Because NCE is symmetric (i.e., also asymmetric), and we have already proved that VBL is asymmetric. So NCE+VBL is still asymmetric and therefore noise-tolerant.

## 4 Experiments

In this section, we provide extensive experiments to evaluate the effectiveness of variation-bounded losses with various types of label noise. Detailed experiment settings are included in the Appendix.

**Datasets.** We consider several datasets with various types of label noise, including CIFAR-10 / 100 (Krizhevsky et al., 2009) for symmetric, asymmetric, instance-dependent (Xia et al., 2020; Chen et al., 2021), and human-annotated (CIFAR-10N / 100N (Wei et al., 2021)) noise; WebVision (Li et al., 2017), ILSVRC12 (Deng et al., 2009), and Clothing1M (Xiao et al., 2015) for large scale real-world noise.

**Baselines.** We experiment with various state-of-the-art methods, including Cross Entropy (CE); Mean Absolute Error (MAE); Mean Square Error (MSE); Generalized Cross Entropy (GCE) (Zhang & Sabuncu, 2018); Symmetric Cross Entropy (SCE) (Wang et al., 2019); Normalized Cross Entropy (NCE) (Ma et al., 2020); Active Passive Loss (APL) (Ma et al., 2020), including NCE+MAE and NCE+RCE; Asymmetric Loss Functions (ALFs) (Zhou et al., 2021; 2023), including NCE+AUL and NCE+AGCE; Active Negative Loss (ANL) (Ye et al., 2023), including NCE+NNCE and NFL+NNFL.

**Hyperparameter Selection.** For VBL, we can easily calculate a rough range for $a$ using Theorem 3. For example, in the case of VCE on CIFAR-10 with 0.8 symmetric noise, we require $v(L_{\text{VCE}}) = \frac{a}{1+a} \leq \frac{1-\eta_{\mathbf{x}}}{\max_{k \neq y} \eta_{\mathbf{x},k}} = \frac{0.2}{0.8/9}$, i.e., $a \geq 0.8$. Based on this calculation, we proceed to search for hyperparameters through empirical testing.

Table 1: Last epoch test accuracies (%) of different methods on CIFAR-10 / 100 symmetric noise ($\eta \in [0.2, 0.4, 0.6, 0.8]$) and asymmetric noise ($\eta \in [0.1, 0.2, 0.3, 0.4]$). The results "mean±std" are reported over 3 random trials and the top-3 best results are **boldfaced**.

| CIFAR10 | Clean | Symmetric | | | | Asymmetric | | | |
|---|---|---|---|---|---|---|---|---|---|
| | | 0.2 | 0.4 | 0.6 | 0.8 | 0.1 | 0.2 | 0.3 | 0.4 |
| CE | 90.50±0.22 | 75.21±0.39 | 58.05±0.53 | 58.05±0.53 | 19.74±0.40 | 86.85±0.15 | 83.05±0.35 | 78.37±0.61 | 73.85±0.07 |
| MAE | 90.02±0.16 | 88.03±0.22 | 82.03±3.63 | 78.54±0.32 | 44.45±6.49 | 89.12±0.15 | 77.20±4.45 | 65.61±2.93 | 57.86±1.23 |
| MSE | 82.65±0.11 | 79.16±0.40 | 72.76±0.39 | 61.82±0.19 | 39.40±0.50 | 81.70±0.29 | 79.62±0.64 | 76.44±0.56 | 71.56±0.25 |
| GCE | 89.36±0.19 | 89.36±0.19 | 82.19±0.84 | 68.01±0.40 | 46.61±0.39 | 88.41±0.20 | 85.72±0.22 | 79.49±0.20 | 73.36±0.53 |
| SCE | 91.51±0.24 | 87.65±0.36 | 79.73±0.29 | 61.79±0.72 | 28.01±0.92 | 89.54±0.33 | 85.94±0.38 | 80.50±0.09 | 74.33±0.56 |
| NCE | 75.48±0.37 | 73.22±0.35 | 69.37±0.22 | 62.47±0.85 | 41.20±1.25 | 74.11±0.24 | 72.20±0.38 | 70.14±0.27 | 65.33±0.40 |
| NCE+MAE | 89.07±0.24 | 87.13±0.11 | 83.38±0.14 | 76.71±0.88 | 45.76±0.99 | 88.07±0.25 | 86.31±0.15 | 83.07±0.27 | 75.71±0.14 |
| NCE+RCE | 90.80±0.06 | 88.93±0.04 | 85.89±0.31 | 79.89±0.25 | 54.99±2.13 | 90.04±0.17 | 88.62±0.29 | 85.07±0.27 | 77.94±0.21 |
| NCE+AUL | 91.17±0.18 | 89.00±0.58 | 86.05±0.30 | 79.22±0.22 | 56.24±0.94 | 90.06±0.16 | 88.19±0.07 | 84.83±0.47 | 77.60±0.16 |
| NCE+AGCE | 91.01±0.20 | 88.91±0.38 | 86.16±0.38 | 79.93±0.33 | 43.82±1.91 | **90.29±0.05** | 88.49±0.28 | 85.21±0.59 | **78.47±1.05** |
| NCE+NNCE | 91.74±0.18 | 89.68±0.29 | 87.16±0.16 | 81.28±0.63 | 62.28±1.10 | **90.66±0.16** | **89.09±0.21** | 85.49±0.49 | 77.99±0.40 |
| **VCE** | 91.63±0.09 | **90.31±0.23** | **87.62±0.19** | 82.22±0.34 | **64.25±0.41** | **90.69±0.23** | **89.46±0.14** | 85.29±0.15 | 75.48±3.85 |
| **VEL** | 91.72±0.18 | **90.12±0.11** | **87.76±0.17** | **82.47±0.43** | **64.03±0.89** | 90.02±0.15 | **89.40±0.18** | **87.16±0.60** | **84.20±0.17** |
| **VMSE** | 91.73±0.16 | 89.95±0.20 | 87.43±0.12 | 82.23±0.49 | **64.85±2.41** | 89.98±0.09 | 88.73±0.22 | **85.35±0.14** | 80.12±0.46 |

| CIFAR100 | Clean | Symmetric | | | | Asymmetric | | | |
|---|---|---|---|---|---|---|---|---|---|
| | | 0.2 | 0.4 | 0.6 | 0.8 | 0.1 | 0.2 | 0.3 | 0.4 |
| CE | 70.93±0.77 | 56.47±1.34 | 39.68±0.77 | 22.64±0.53 | 7.82±0.33 | 64.14±1.01 | 58.67±0.45 | 50.44±1.16 | 41.51±0.12 |
| MAE | 6.75±1.29 | 3.82±0.06 | 3.66±1.12 | 3.09±0.52 | 2.55±0.28 | 5.29±1.31 | 4.34±1.04 | 3.64±1.31 | 3.15±0.83 |
| MSE | 37.40±0.43 | 29.32±0.91 | 20.99±0.58 | 13.00±0.99 | 6.34±0.46 | 33.81±0.44 | 29.05±0.34 | 23.41±0.25 | 18.05±0.38 |
| GCE | 61.73±1.30 | 60.58±2.51 | 57.35±0.91 | 46.15±1.10 | 20.33±0.31 | 62.01±1.11 | 59.19±1.36 | 53.35±0.65 | 40.92±0.21 |
| SCE | 70.57±0.93 | 55.50±0.35 | 40.13±1.48 | 22.23±1.29 | 7.84±0.56 | 64.51±0.45 | 57.84±0.57 | 49.66±0.48 | 41.58±0.87 |
| NCE | 29.95±0.56 | 25.43±0.91 | 20.26±0.25 | 14.66±1.04 | 8.82±0.47 | 27.16±1.01 | 26.67±0.73 | 23.83±0.29 | 20.83±1.08 |
| NCE+MAE | 67.49±0.31 | 52.79±0.39 | 35.62±0.22 | 19.65±0.95 | 7.04±0.64 | 60.16±0.50 | 52.96±0.53 | 44.20±0.16 | 36.25±0.60 |
| NCE+RCE | 68.07±0.70 | 64.57±0.16 | 58.48±0.51 | 46.73±1.00 | 26.94±1.29 | 66.74±0.30 | 62.82±0.57 | 55.86±0.40 | 41.50±0.39 |
| NCE+AUL | 69.95±0.33 | 65.45±0.49 | 56.37±0.12 | 38.68±0.75 | 12.95±0.37 | 66.41±0.15 | 57.39±0.34 | 48.20±0.19 | 38.41±0.52 |
| NCE+AGCE | 69.05±0.36 | 65.61±0.27 | 59.40±0.34 | 47.66±0.49 | 26.14±0.01 | 66.96±0.45 | 64.08±0.44 | 57.17±0.33 | 44.62±1.04 |
| NCE+NNCE | 70.26±0.15 | 66.93±0.09 | 61.58±0.33 | 52.09±0.58 | **28.01±1.06** | 68.60±0.41 | 65.96±0.18 | 60.57±0.07 | 45.73±0.74 |
| **NCE+VCE** | 72.06±0.70 | **69.37±0.21** | **64.87±0.69** | **56.71±0.44** | **33.61±0.91** | **70.29±0.65** | **68.09±0.22** | **65.06±0.37** | **55.82±0.99** |
| **NCE+VEL** | 72.85±0.56 | **70.21±0.47** | **65.37±0.37** | **57.82±0.26** | **35.06±0.98** | **70.56±0.28** | **68.52±0.11** | **65.38±0.14** | 53.31±0.55 |
| **NCE+VMSE** | 70.87±0.16 | **68.22±0.49** | **64.32±0.46** | **56.27±0.46** | 21.60±1.82 | **69.42±0.18** | **67.34±0.10** | **64.67±0.42** | **57.60±0.23** |

Table 2: Last epoch test accuracies (%) of different methods on CIFAR-10 / 100 instance-dependent noise ($\eta \in [0.2, 0.4, 0.6]$). The results "mean±std" are reported over 3 random trials and the top-3 best results are **boldfaced**.

| CIFAR-10 | Instance-Dependent Noise | | | CIFAR-100 | Instance-Dependent Noise | | |
|---|---|---|---|---|---|---|---|
| | 0.2 | 0.4 | 0.6 | | 0.2 | 0.4 | 0.6 |
| CE | 75.38±0.19 | 57.63±0.27 | 37.97±0.36 | CE | 57.02±0.54 | 40.91±2.05 | 24.49±0.86 |
| GCE | 86.66±0.14 | 79.99±0.23 | 51.90±0.13 | GCE | 61.43±2.24 | 57.07±1.04 | 42.40±0.52 |
| SCE | 86.65±0.27 | 74.54±0.34 | 49.83±0.40 | SCE | 56.32±0.27 | 39.82±1.43 | 23.19±0.87 |
| NCE+RCE | 89.06±0.26 | 85.11±0.28 | 71.27±0.66 | NCE+RCE | 64.33±0.46 | 57.53±0.84 | 40.36±0.35 |
| NCE+AGCE | 88.95±0.07 | 85.30±0.23 | 71.49±0.34 | NCE+AGCE | 65.18±0.17 | 57.89±0.57 | 43.04±0.29 |
| NCE+NNCE | **89.71±0.35** | 85.74±0.15 | 69.83±0.38 | NCE+NNCE | 66.89±0.53 | 60.88±0.35 | 48.12±0.48 |
| **VCE** | 89.53±0.42 | **86.77±0.12** | **75.98±0.87** | **NCE+VCE** | 69.33±0.24 | 64.54±0.46 | 54.04±0.58 |
| **VEL** | 89.74±0.24 | **86.82±0.33** | 74.64±0.85 | **NCE+VEL** | 70.38±0.16 | 65.06±0.26 | 55.20±0.32 |
| **VMSE** | 89.95±0.08 | 87.33±0.12 | 74.27±0.42 | **NCE+VMSE** | 67.51±0.12 | 63.27±0.73 | 52.29±0.85 |

For NCE+VBL, we can similarly estimate a rough range for $a$ using the same method. After determining this range, we empirically tune the hyperparameters $\alpha$, $\beta$, and $a$. To simplify the process, in most cases, we fix $\alpha = 5$ and $\beta = 1$, and then search only for $a$. In rare cases where $\alpha = 5$ does not work very well, we also search for $\alpha$. Since $\beta$ is always fixed at 1, there are only two parameters to adjust in practice. In our experiments, this method of parameter tuning consistently delivers good performance.

Table 3: Last epoch test accuracies (%) of different methods on CIFAR-10N / 100N human-annotated noise. The results "mean±std" are reported over 3 random trials and the top-3 best results are **boldfaced**.

| CIFAR-10N | Human | | | | | CIFAR-100N | Human |
|---|---|---|---|---|---|---|---|
| | Aggregate | Random 1 | Random 2 | Random 3 | Worst | | Noisy |
| CE | 85.09±0.30 | 79.09±0.28 | 78.59±0.42 | 78.39±0.50 | 61.43±0.52 | CE | 48.63±0.53 |
| GCE | 87.38±0.07 | 85.87±0.27 | 85.43±0.13 | 85.51±0.15 | 75.19±0.23 | GCE | 50.97±0.60 |
| SCE | 88.48±0.26 | 85.65±0.30 | 85.71±0.19 | 85.87±0.13 | 73.65±0.29 | SCE | 48.52±0.11 |
| NCE+RCE | 89.17±0.28 | 87.62±0.34 | 87.66±0.12 | 87.70±0.18 | 79.74±0.09 | NCE+RCE | 54.27±0.09 |
| NCE+AGCE | 89.27±0.28 | 87.92±0.02 | 87.61±0.20 | 87.62±0.16 | 79.91±0.37 | NCE+AGCE | 55.96±0.20 |
| NCE+NNCE | 89.66±0.12 | 88.68±0.13 | 88.19±0.08 | 88.24±0.15 | 80.23±0.28 | NCE+NNCE | 56.37±0.42 |
| **VCE** | **90.05±0.14** | **89.08±0.09** | **88.62±0.16** | **88.82±0.21** | **81.85±0.35** | **NCE+VCE** | **58.75±0.32** |
| **VEL** | **90.10±0.14** | **88.98±0.19** | **88.76±0.23** | **89.05±0.14** | **81.80±0.07** | **NCE+VEL** | **59.54±0.14** |
| **VMSE** | **90.08±0.18** | **89.13±0.05** | **88.96±0.21** | **88.84±0.16** | **81.89±0.30** | **NCE+VMSE** | **59.68±0.46** |

Table 4: Last epoch test accuracies (%) of different methods on CIFAR-100 symmetric noise ($\eta \in [0.2, 0.4, 0.6, 0.8]$). The results "mean±std" are reported over 3 random trials and the best results are **boldfaced**.

| Method | CIFAR-10 Symmetric Noise | | | | CIFAR-100 Symmetric Noise | | | |
|---|---|---|---|---|---|---|---|---|
| | 0.2 | 0.4 | 0.6 | 0.8 | 0.2 | 0.4 | 0.6 | 0.8 |
| CE | 75.21±0.39 | 58.05±0.53 | 58.05±0.53 | 19.74±0.40 | 56.47±1.34 | 39.68±0.77 | 22.64±0.53 | 7.82±0.33 |
| NCE | 73.22±0.35 | 69.37±0.22 | 62.47±0.85 | 41.20±1.25 | 25.43±0.91 | 20.26±0.25 | 14.66±1.04 | 8.82±0.47 |
| VCE | **90.31±0.23** | 87.62±0.19 | 82.22±0.34 | 64.25±0.41 | 65.37±0.21 | 59.88±0.49 | 49.47±1.16 | 30.80±0.83 |
| NCE+VCE | 90.13±0.25 | **87.71±0.12** | **82.59±0.08** | **64.92±1.87** | **69.37±0.21** | **64.87±0.69** | **56.71±0.44** | **33.61±0.91** |

## 4.1 EVALUATION ON BENCHMARK DATASETS

In this subsection, we evaluate our variation-bounded losses on benchmark datasets CIFAR-10 / 100 (Krizhevsky et al., 2009) following the same setting in (Ma et al., 2020; Zhou et al., 2021; Ye et al., 2023). We use an 8-layer CNN and a ResNet-34 (LeCun et al., 1989; He et al., 2016) for CIFAR-10 and CIFAR-100, respectively.

**Results.** Table 1 and 2 showcase the test accuracy of different methods under various synthetic noise, including symmetric, asymmetric, and instance-dependent label noise. Notably, our introduced variation-bounded losses, VCE, VEL, and VMSE, exhibit exceptional performance, consistently ranking among the top-3 in most cases. Furthermore, for clean labels, our variation-bounded losses continue to demonstrate superior fitting abilities, outperforming GCE, NCE+RCE, NCE+AGCE, etc. Notably, under 0.8 symmetric, 0.4 asymmetric, and 0.6 instance-dependent noise, our method improves accuracy by 2% to 12% over previous state-of-the-art methods. Furthermore, we conduct experiments with human-annotated label noise using the CIFAR-10N / 100N datasets, as shown in Table 3. As can be seen, our variation-bounded losses achieve top-3 performance across all human-annotated cases, highlighting the excellent performance of our method in such scenarios. These results demonstrate that our method significantly surpasses the latest benchmarks.

**More Discussion about NCE+VBL.** We can observe that VBL alone often outperforms state-of-the-art methods in most scenarios on CIFAR-10. Therefore, for CIFAR-10, we opt to use VBL alone to better highlight the superiority of our method, even though it puts us at a disadvantage compared to methods that combine NCE. Since CIFAR-100 is a more challenging dataset, the improvement from incorporating NCE is more pronounced. If we do not combine with NCE, we cannot completely exceed the previous sota methods that also utilize NCE. Therefore, we opt to combine NCE with VBL on CIFAR-100 to surpass them.

In addition, we present more results using VCE alone and NCE+VCE, as shown in Table 4. As can be seen, a simple modification yields a significant improvement for VCE over vanilla CE. Furthermore, on CIFAR-10, there is little difference in performance between VCE and NCE+VCE, and on CIFAR-100, NCE+VCE achieves better results.

**Visualization.** We conduct a further analysis to compare the robustness of variation-bounded losses and vanilla CE in learning representations. We train models on CIFAR-10 with 0.4 symmetric

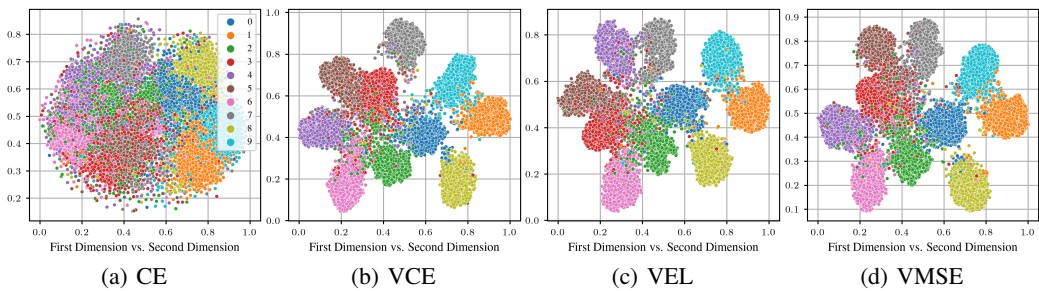

| (a) CE | (b) VCE | (c) VEL | (d) VMSE |

Figure 2: Visualizations of 2D embeddings for learned features on CIFAR-10 with 0.4 symmetric noise by t-SNE (Van der Maaten & Hinton, 2008).

Table 5: Last epoch test accuracies (%) of different methods on WebVision, ILSVRC12, and Clothing1M. Baseline results for WebVision and ILSVRC12 are taken from (Ye et al., 2023) with the same setting. The top-2 best results are **boldfaced**.

| Method | CE | GCE | SCE | NCE+RCE | NCE+AGCE | NCE+NNCE | NFL+NNFL | VCE | NCE+VCE |
|--------|-----|-----|-----|---------|----------|----------|----------|-----|---------|
| **WebVision** | 61.20 | 59.44 | 68.00 | 64.92 | 63.92 | 67.44 | 68.32 | **70.12** | **69.36** |
| **ILSVRC12** | 58.64 | 56.56 | 62.60 | 62.40 | 60.76 | 65.00 | 65.56 | **66.12** | **66.16** |
| **Clothing1M** | 67.38 | 69.03 | 67.40 | 68.67 | 67.52 | 69.75 | 69.90 | **70.23** | **70.49** |

noise and extract the learned features from the test set. These features are then projected into 2D embeddings using t-SNE (Van der Maaten & Hinton, 2008), as depicted in Figure 2. As can be seen, embeddings generated by CE exhibit evident overfitting to label noise, as evidenced by the blending of embeddings from distinct classes. In contrast, embeddings generated by variation-bounded losses consistently form clear, well-separated clusters. This demonstrates their superior capability to learn robust and distinct representations under label noise.

### 4.2 Evaluation on Real-World Datasets

We perform experiments on massively real-world noisy datasets, including WebVision (Li et al., 2017), ILSVRC12 (ImageNet) (Deng et al., 2009), and Clothing1M (Xiao et al., 2015), following the same setting in (Ma et al., 2020; Ye et al., 2023). We train a ResNet-50 on the WebVision noisy training set and subsequently evaluate the trained model on both ILSVRC2012 and WebVision clean validation sets. For Clothing1M, we use a ResNet-50 pre-trained on ImageNet similar to (Xiao et al., 2015; Ye et al., 2023). We train the model on the noisy training set and subsequently evaluate it on the clean test set.

**Results.** In Table 5, we showcase the accuracies achieved on WebVision, ILSVRC12, and Clothing1M by various leading methods. Notably, our variation-bounded loss, outperforms others, achieving the highest results on all real-world datasets. Specifically, our VCE and NCE+VCE surpasses vanilla CE by approximately 8.9% on WebVision, 7.5% on ILSVRC12, and 4.1% on Clothing1M. For Clothing1M, we use a pre-trained ResNet-50, resulting in relatively smaller differences between methods, but our methods still achieves the best accuracy. These results underline the robustness and effectiveness of variation-bounded losses in real-world scenarios.

## 5 Conclusion

In this paper, we introduce the *Variation Ratio* to measure the robustness of loss functions. Beside on this, we propose a new family of robust loss functions, namely *Variation-Bounded Losses* (VBL). We prove that a smaller variation ratio represents a better robustness and show that variation-bounded losses have a more relaxed condition than symmetric losses. Our concise robust loss functions has shown positive results for mitigating label noise in a variety of label noise types. We believe that they can be widely applied in scenarios where obtaining accurate annotations is challenging. Additionally, the variation ratio will serve as a valuable tool for designing more effective robust loss functions.

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

# APPENDIX

## A PROOFS

### A.1 PROOF FOR LEMMA 1

**Lemma 1.** *Consider a monotone decreasing basic loss $\ell_{active}$, for a loss function $L(\mathbf{u}, y) = c \cdot \ell_{active}(u_y)$, we have*

$$|\sum_{k=1}^{K} L(\mathbf{u}, k) - \sum_{k=1}^{K} L(\mathbf{v}, k)| \leq v(L) - 1. \tag{A.1}$$

*where $c = \frac{1}{\min_u |\nabla \ell_{active}|}$ is a normalization constant.*

*Proof.* For brevity, we set $\ell = \ell_{active}$. For any $\mathbf{u}$, we have $\sum_{k=1}^{K} \ell(0) - \sum_{k=1}^{K} u_k \max_u |\nabla \ell| \leq \sum_{k=1}^{K} \ell(u_k) \leq \sum_{k=1}^{K} \ell(0) - \sum_{k=1}^{K} u_k \min_u |\nabla \ell|$, this is

$$K \cdot \ell(0) - \max_u |\nabla \ell| \leq \sum_{k=1}^{K} \ell(u_k) \leq K \cdot \ell(0) - \min_u |\nabla \ell|. \tag{A.2}$$

Hence, we have $|\sum_{k=1}^{K} \ell(u_k) - \sum_{k=1}^{K} \ell(v_k)| \leq \max_u |\nabla \ell| - \min_u |\nabla \ell|$ and $|\sum_{k=1}^{K} L(\mathbf{u}, k) - \sum_{k=1}^{K} L(\mathbf{v}, k)| \leq v(L) - 1$. $\qquad \square$

### A.2 PROOF FOR THEOREM 1

**Theorem 1** (Excess Risk Bound under Symmetric Noise). *In a multi-class classification problem, if the loss function $L \in \mathcal{L}$ satisfies $|\sum_{k=1}^{K} L(\mathbf{u}, k) - \sum_{k=1}^{K} L(\mathbf{v}, k)| \leq v(L) - 1$, then for symmetric noise satisfying $\eta < 1 - \frac{1}{K}$, the excess risk bound for $f$ can be expressed as*

$$\mathcal{R}_L(f_\eta^*) - \mathcal{R}_L(f^*) \leq c(v(L) - 1), \tag{A.3}$$

*where $c = \frac{\eta}{(1-\eta)K - 1}$ is a constant, $f_\eta^*$ and $f^*$ denote the global minimum of $\mathcal{R}_L^\eta(f)$ and $\mathcal{R}_L(f)$, respectively.*

*Proof.* For symmetric noise, we have

$$R_L^\eta(f^*) = \mathbb{E}_{\mathbf{x}, y}\left[(1 - \eta)L(f^*(\mathbf{x}), y) + \frac{\eta}{K - 1}\sum_{k \neq y} L(f^*(\mathbf{x}), k)\right]$$

$$= (1 - \frac{\eta K}{K - 1})R_L(f^*) + \frac{\eta}{K - 1}\mathbb{E}_{\mathbf{x}, y}\left[\sum_{k=1}^{K} L(f^*(\mathbf{x}), k)\right] \tag{A.4}$$

Similarly, we can obtain

$$R_L^\eta(f_\eta^*) = (1 - \frac{\eta K}{K - 1})R_L(f_\eta^*) + \frac{\eta}{K - 1}\mathbb{E}_{\mathbf{x}, y}\left[\sum_{k=1}^{K} L(f_\eta^*(\mathbf{x}), k)\right] \tag{A.5}$$

Since $f_\eta^* = \arg\min_u R_L^\eta(f)$, and $f^* = \arg\min_u R_L(f)$, we have

$$R_L^\eta(f_\eta^*) - R_L^\eta(f^*)$$

$$= (1 - \frac{\eta K}{K - 1})(R_L(f_\eta^*) - R_L(f^*)) + \frac{\eta}{K - 1}\mathbb{E}_{\mathbf{x}, y}[\sum_{k=1}^{K} L(f_\eta^*(\mathbf{x}), k) - \sum_{k=1}^{K} L(f^*(\mathbf{x}), k)] \leq 0$$

$$\Rightarrow R_L(f_\eta^*) - R_L(f^*) \leq \frac{\eta}{(1 - \eta)k - 1}(v(L) - 1)$$

$$\tag{A.6}$$

where we have used the fact that $1 - \frac{\eta k}{k - 1} > 0$. $\qquad \square$

### A.3 PROOF FOR THEOREM 2

**Theorem 2** (Excess Risk Bound under Asymmetric and Instance-Dependent Noise).*In a multi-class classification problem, if the loss function $L \in \mathcal{L}$ satisfies $|\sum_{k=1}^{K} L(\mathbf{u}, k) - \sum_{k=1}^{K} L(\mathbf{v}, k)| \leq v(L) - 1$, then for label noise $1 - \eta_{\mathbf{x}} > \max_{k \neq y} \eta_{\mathbf{x},k}, \forall \mathbf{x}$, if $\mathcal{R}_L(f^*)$ is minimum, the excess risk bound for $f$ can be expressed as*

$$\mathcal{R}_L(f_\eta^*) - \mathcal{R}_L(f^*) \leq (1 + \frac{c}{a})(v(L) - 1), \tag{A.7}$$

*where $c = \mathbb{E}_{\mathcal{D}}(1 - \eta_{\mathbf{x}})$ and $a = \min_{\mathbf{x},k}(1 - \eta_{\mathbf{x}} - \eta_{\mathbf{x},k})$ are constants, $f_\eta^*$ and $f^*$ denote the global minimum of $\mathcal{R}_L^\eta(f)$ and $\mathcal{R}_L(f)$, respectively. For asymmetric noise, $\eta_{\mathbf{x}} = \eta_y$, and for instance-dependent noise, $\eta_{\mathbf{x}} = \eta_{\mathbf{x}}$.*

*Proof.* For asymmetric and instance-dependent noise, we have

$$R_L^\eta(f) = \mathbb{E}_{\mathcal{D}}\left[(1 - \eta_{\mathbf{x}}) L(f(\mathbf{x}), y)\right] + \mathbb{E}_{\mathcal{D}}[\sum_{k \neq y} \eta_{\mathbf{x},k} L(f(\mathbf{x}), k)]$$

$$= \mathbb{E}_{\mathcal{D}}\left[(1 - \eta_{\mathbf{x}})\left(\sum_{k=1}^{K} L(f(\mathbf{x}), y) - \sum_{k \neq y} L(f(\mathbf{x}), k)\right)\right] + \mathbb{E}_{\mathcal{D}}\left[\sum_{k \neq y} \eta_{\mathbf{x},k} L(f(\mathbf{x}), k)\right]$$

$$= \sum_{k=1}^{K} L(f(\mathbf{x}), y)\mathbb{E}_{\mathcal{D}}(1 - \eta_{\mathbf{x}}) - \mathbb{E}_{\mathcal{D}}\left[\sum_{k \neq y}(1 - \eta_{\mathbf{x}} - \eta_{\mathbf{x},k})L(f(\mathbf{x}), k)\right]$$

hence,

$$\left(R_L^\eta(f^*) - R_L^\eta(f_\eta^*)\right) = (\sum_{k=1}^{K} L(f^*(\mathbf{x}), y) - \sum_{k=1}^{K} L(f_\eta^*(\mathbf{x}), y))\mathbb{E}_{\mathcal{D}}(1 - \eta_{\mathbf{x}}) +$$

$$\mathbb{E}_{\mathcal{D}} \sum_{k \neq y}(1 - \eta_{\mathbf{x}} - \eta_{\mathbf{x},k})\left[L(f_\eta^*(\mathbf{x}), k) - L(f^*(\mathbf{x}), k)\right]$$

According to the assumption $R_L(f^*)$ is minimum, we have $L(f^*(\mathbf{x}), y)$ is minimum then $L(f^*(\mathbf{x}), k)$ is maximum where $k \neq y$. Since $L(f_\eta^*(\mathbf{x}), k) - L(f^*(\mathbf{x}), k) \leq 0$ where $k \neq y$, the second term on the right of the inequality is a non-positive value. And $R_L^\eta(f^*) - R_L^\eta(f_\eta^*) \geq 0$. So we have

$$\left|\mathbb{E}_{\mathcal{D}} \sum_{k \neq y}(1 - \eta_{\mathbf{x}} - \eta_{\mathbf{x},k})\left(L(f_\eta^*(\mathbf{x}), k) - L(f^*(\mathbf{x}), k)\right)\right| \leq c(v(L) - 1),$$

where $c = \mathbb{E}_{\mathcal{D}}(1 - \eta_{\mathbf{x}})$.

Let $a = \min_{\mathbf{x},k}(1 - \eta_{\mathbf{x}} - \eta_{\mathbf{x},k})$, we have $\left|\mathbb{E}_{\mathcal{D}} \sum_{k \neq y}\left(L(f_\eta^*(\mathbf{x}), k) - L(f^*(\mathbf{x}), k)\right)\right| \leq \frac{c(v(L)-1)}{a}$. Note that$|\sum_k \left(L(f_\eta^*(\mathbf{x}), k) - L(f^*(\mathbf{x}), k)\right)| \leq v(L) - 1$, then we obtain

$$\left|\mathbb{E}_{\mathcal{D}}\left(L(f_\eta^*(\mathbf{x}), y) - L(f^*(\mathbf{x}), y)\right)\right| \leq (v(L) - 1) + \frac{c(v(L) - 1)}{a},$$

that is, $\mathcal{R}_L(f_\eta^*) - \mathcal{R}_L(f^*) \leq (1 + \frac{c}{a})(v(L) - 1)$. □

### A.4 PROOF FOR THEOREM 3

**Theorem 3** .*On the given weights $w_1, \ldots, w_k \geq 0$, where $\exists t \in [K]$ and $w_t > \max_{i \neq t} w_i$, a monotone decreasing loss function $L(\mathbf{u}, k) = \ell_{active}(u_k)$ is asymmetric if (1) $\frac{\partial^2 \ell_{active}(u)}{\partial u^2} \leq 0$ or (2) $v(L) \leq \frac{w_t}{w_i}$ for any $i \neq t$.*

*Proof.* According to (Zhou et al., 2021), for any $w_1 > w_2 \geq 0$, if $\ell$ satisfies $w_1\ell(u_1) + w_2\ell(u_2) = w_1\ell(u_1 + u_2) + w_2\ell(0)$, and the equality holds only if $u_2 = 0$, then $L$ is completely asymmetric.

This is

$$w_1(\ell(u_1) - \ell(u_1 + u_2)) \geq w_2(\ell(0) - \ell(u_2))$$

$$\Rightarrow w_1 \frac{(\ell(u_1) - \ell(u_1 + u_2))}{u_2} \geq w_2 \frac{(\ell(0) - \ell(u_2))}{u_2}$$

If $\frac{\partial^2 \ell(u_k)}{\partial u_k^2} \leq 0$, we have $\ell(u_1) - \ell(u_1 + u_2) \geq \ell(0) - \ell(u_2)$, because $\nabla\ell(x + u_1) < \nabla\ell(x)$, thus established. In other cases, according to Lagrange's mean value theorem, we have

$$w_1(|\nabla\ell(\xi_1)|) \geq w_2(|\nabla\ell(\xi_2)|) \tag{A.8}$$

where $\xi_1 \in [u_1, u_1 + u_2]; \xi_2 \in [0, u_2]$. we have $\frac{w_1}{w_2} \geq \frac{|\nabla\ell(\xi_2)|}{|\nabla\ell(\xi_1)|}$, if $\frac{w_1}{w_2} \geq \frac{\max_u |\nabla\ell|}{\min_u |\nabla\ell|} = v(L)$, E.q. A.8 is true.

$\square$

## A.5    PROOF FOR COROLLARY 1

**Corollary 1** .*On the given weights $w_1, \ldots, w_k$, where $w_m > w_n$, and $w_n = \max_{k \neq m} w_k$. The loss function $L_p(\mathbf{u}, y) = \frac{1}{K}\|a \cdot \mathbf{e}_y - \mathbf{u}\|_p^p = \frac{1}{K}[(a - u_y)^p + \sum_{k \neq y} u_k^p]$, where $p > 0$ and $a \geq 1$ are hyper-parameters, is asymmetric if and only if $\frac{w_m}{w_n} \geq [v(L_p) + \frac{\sum_{k \neq m} \frac{w_k}{w_n}}{(a-1)^{p-1}}] \cdot \mathbb{I}(p > 1) + \mathbb{I}(p \leq 1)$.*

*Proof.* If $L_p(u, i)$ is asymmetric, then for $w_m > w_n \geq 0$, let $u_i = 0$, $i \neq m, n$, we have $\sum_{i=1}^k w_i L(\mathbf{u}, i) \geq \sum_{i=1}^k w_i L(\mathbf{e}_m, i)$ always holds, i.e.,

$$w_m[(a - u_m)^p + u_n] + w_n[(a - u_n)^p + u_m^p] + \sum_{i \neq m,n} w_i[a^p + u_m^p + u_n^p]$$

$$\geq w_m[(a-1)^p] + w_n(a^p + 1) + \sum_{i \neq m} w_i(a^p + 1)$$

so we have

$$\frac{w_m}{w_n} \geq \sup_{\substack{u_m, u_n \geq 0 \\ u_m + u_n = 1}} \frac{a^p + 1 - (a - u_n)^p - u_m^p + \sum_{i \neq m,n} \frac{w_i}{w_n}(1 - u_m^p - u_n^p)}{(a - u_m)^p + u_n^p - (a-1)^p}$$

$$= \frac{a^{p-1} + 1 + \sum_{i \neq m} \frac{w_i}{w_n}}{(a-1)^{p-1}} \cdot \mathbb{I}(p > 1) + \mathbb{I}(p \leq 1)$$

$$= \frac{a^{p-1} + \sum_{i \neq m} \frac{w_i}{w_n}}{(a-1)^{p-1}} \cdot \mathbb{I}(p > 1) + \mathbb{I}(p \leq 1)$$

Let $\mathbf{u}' \in \mathcal{U}$, where $u_m' = u_m + u_n, u_n = 0$, and $u_i' = u_i$ for $i \neq m, n$, then $L$ is asymmetric

$$\Leftrightarrow \sum_{i=1}^k w_i L(\mathbf{u}, i) \geq \sum_{i=1}^k w_i L(\mathbf{u}', i)$$

$$\Leftrightarrow \frac{w_m}{w_k} \geq \sup_{\substack{u_m, u_n \geq 0 \\ u_m + u_k = 1}} \frac{a^p + (u_m + u_k)^p - (a - u_k)^p - u_m^p + \sum_{i \neq m,k} \frac{w_i}{w_k}[(u_m + u_k)^p - u_m^p - u_k^2]}{(a - u_m)^p + u_k^p - (a - u_m - u_k)^p}$$

$$\Leftrightarrow \frac{w_m}{w_k} \geq \frac{a^{p-1} + \sum_{i \neq m} \frac{w_i}{w_k}}{(a-1)^{p-1}} \cdot \mathbb{I}(p > 1) + \mathbb{I}(p \leq 1)$$

On the other hand, $\frac{w_m}{w_n} \geq \frac{a^{p-1} + \sum_{i \neq m} \frac{w_i}{w_n}}{(a-1)^{p-1}} \cdot \mathbb{I}(p > 1) + \mathbb{I}(p \leq 1)$. Then for any $k \neq m$, we have $\frac{w_m}{w_k} \geq \frac{a^{p-1} + \sum_{i \neq m} \frac{w_i}{w_k}}{(a-1)^{p-1}} \cdot \mathbb{I}(p > 1) + \mathbb{I}(p \leq 1)$, where $v(L_p) = \frac{a^{p-1}}{(a-1)^{p-1}}$ and $\frac{a^{p-1} + \sum_{i \neq m} \frac{w_i}{w_k}}{(a-1)^{p-1}} = v(L_p) + \frac{\sum_{k \neq m} \frac{w_k}{w_n}}{(a-1)^{p-1}}$.

$\square$

## B EXPERIMENTS

### B.1 EVALUATION ON BENCHMARK DATASETS

**Noise Generation.**  We follow the approach of the previous work (Ye et al., 2023) to experiment with two types of synthetic label noise: symmetric noise and asymmetric noise. In the case of symmetric label noise, we intentionally corrupt the training labels by randomly flipping labels within each class to incorrect labels in other classes. As for asymmetric label noise, we flip the labels within a specific sets of classes: For CIFAR-10, the flips occur from TRUCK → AUTOMOBILE, BIRD → AIRPLANE, DEER → HORSE, and CAT ↔ DOG. For CIFAR-100, the 100 classes are grouped into 20 super-classes, each containing 5 sub-classes, and we flip the labels within the same super-class into the next. For instance-dependent noise, we follow the approach in PDN (Xia et al., 2020) for generating label noise.

**Experimental Setting.**  We follow the experimental settings in (Ma et al., 2020; Zhou et al., 2021; Ye et al., 2023): An 8-layer CNN is used for CIFAR-10 and a ResNet-34 (LeCun et al., 1989; He et al., 2016) for CIFAR-100. The networks are trained for 120 and 200 epochs for CIFAR-10 and CIFAR-100 with batch size 128. We use the SGD optimizer with momentum 0.9 and L1 weight decay $5 \times 10^{-5}$ and $5 \times 10^{-6}$ for CIFAR-10 and CIFAR-100. The learning rate is set to 0.01 for CIFAR-10 and 0.1 for CIFAR-100 with cosine annealing. Typical data augmentations including random shift and horizontal flip are applied.

**Parameters Setting.**  For baselines, we use the same parameter settings in (Ma et al., 2020; Zhou et al., 2021; Ye et al., 2023), which match their best parameters. For our VBL, the parameter settings can be found in Table 6.

Table 6: Parameters $(\alpha, \beta, a)$ for $\alpha$NCE $+ \beta$VBL. $\alpha = 0$ denotes using VBL only. Parameters for clean label are the same as those for symmetric noise.

| Parameter | CIFAR-10 | | | | CIFAR-100 | | | |
|---|---|---|---|---|---|---|---|---|
| | Symmetric | Asymmetric | Dependent | Human | Symmetric | Asymmetric | Dependent | Human |
| NCE+VCE | (0, 10, 4) | (0, 1, 0.6) | (0, 5, 3.5) | (0, 2, 1.4) | (5, 1, 0.2) | (5, 1, 0.9) | (5, 1, 0.4) | (5, 1, 0.9) |
| NCE+VEL | (0, 10, 1.4) | (0, 1, 25) | (0, 5, 1.5) | (0, 5, 1.5) | (5, 1, 10) | (5, 1, 2) | (5, 1, 7) | (5, 1, 1.5) |
| NCE+VMSE | (0, 1, 30) | (0, 1, 2) | (0, 1, 15) | (0, 1, 9) | (5, 1, 20) | (5, 1, 30) | (5, 1, 15) | (5, 1, 15) |

### B.2 EVALUATION ON REAL-WORLD DATASETS

**Experiment and Parameter Setting for WebVision / ILSVRC12.**  For WebVision, we follow the same setting in (Ma et al., 2020; Ye et al., 2023): We use the "Mini" setting described in (Ma et al., 2020; Ye et al., 2023), which includes the first 50 classes of Webvision. We train a ResNet-50 using SGD for 250 epochs with initial learning rate 0.4, nesterov momentum 0.9 and L1 weight decay $6 \times 10^{-5}$ and batch size 512. The learning rate is multiplied by 0.97 after each epoch of training. All the images are resized to $224 \times 224$. Typical data augmentations including random shift, color jittering, and horizontal flip are applied. We train the model on Webvision and evaluate the trained model on the same 50 concepts on the corresponding WebVision and ILSVRC12 validation sets. For VCE, we set $a = 0.015$. For NCE+VCE, we set $\alpha = 5, \beta = 1, a = 0.02$

**Experiment and Parameter Setting for Clothing1M.**  For Clothing1M, we use ResNet-50 pre-trained on ImageNet similar to (Xiao et al., 2015; Ye et al., 2023). All the images are resized to $224 \times 224$. We use SGD with a momentum of 0.9, a weight decay of $1 \times 10^{-3}$, and batch size of 256. We train the network for 10 epochs with a learning rate of $5 \times 10^{-3}$ and a decay of 0.1 at 5 epochs. Typical data augmentations including random shift and horizontal flip are applied. We use the best parameters for each method. Specifically, for GCE, we set $q = 0.6$. For SCE, we set $A = 3, \alpha = 10, \beta = 1$. For NCE+RCE, we set $\alpha = 10, \beta = 1$. For NCE+AGCE, we set $\alpha = 50, \beta = 0.1, a = 2.5, q = 3$. For NCE+NNCE and NFL+NNFL, we set $\alpha = 5, \beta = 0.1$. For VCE, we set $a = 0.05$. For NCE+VCE, we set $\alpha = 1, \beta = 1, a = 0.6$

Table 7: Last epoch test accuracies (%) of different methods on CIFAR-10 with 0.8 symmetric noise. The results "mean±std" are reported over 3 random trials and the best results are **boldfaced**.

| CIFAR-10 | Symmetric 0.8 | | |
|---|---|---|---|
| | 120 epochs | 200 epochs | 400 epochs |
| GCE ($q = 0.9$) | 46.61±0.39 | 33.39±0.91 | 23.25±0.74 |
| VCE ($a = 4$) | **64.25±0.41** | **63.02±1.34** | 59.55±0.72 |
| VCE ($a = 10$) | 62.73±2.73 | 62.74±2.59 | **61.41±2.02** |

Table 8: Last epoch test accuracies (%) of NCE+VCE with different hyperparameters on CIFAR-100. The results "mean±std" are reported over 3 random trials and the results over the last state-of-the-art method are **boldfaced**.

| CIFAR-100 | Clean | Symmetric 0.8 | Asymmetric 0.4 | Dependent 0.6 | Human |
|---|---|---|---|---|---|
| NCE+NNCE | 70.26±0.15 | 28.01±1.06 | 45.73±0.74 | 48.12±0.48 | 56.37±0.42 |
| ($5, 1, a = 0.2$) | **72.06±0.70** | **33.61±0.91** | 45.35±0.43 | **51.09±1.01** | 55.94±0.23 |
| ($5, 1, a = 0.4$) | **72.48±0.43** | **30.19±0.30** | **51.19±0.33** | **54.04±0.58** | **57.44±0.60** |
| ($5, 1, a = 0.9$) | **71.70±0.11** | 15.51±0.11 | **55.82±0.99** | 46.20±0.43 | **58.75±0.32** |

## B.3 MORE EXPERIMENTS

**More Ablation Experiments about More Epochs.** In Figure 1, we show the accuracy curves of VBL under different variation ratios. We conduct additional experiments of VCE about more epochs on CIFAR-10 with 0.8 Symmetric noise. The results are reported in Table 7.

The results demonstrate two points. (1) As $f_\eta$ converges to the global minimum of $R_L^\eta(f)$, GCE basically loses robustness. Conversely, VCE still retains excellent noise-tolerance. (2) VCE ($a = 10$) achieved a better performance than VCE ($a = 4$) in the case of both training 400 epochs. This shows that a smaller variation ratio provides better robustness. The experimental results consistent with our theory.

**More Ablation Experiments about NCE+VCE.** We conduct more ablation experiments about NCE+VCE on CIFAR-100 with different hyperparameters. The results are reported in Table 8. As can be seen, for fixed hyperparameters ($\alpha = 5, \beta = 1, a = 0.4$), we can also significantly outperform the last state-of-the-art method in all scenarios.

