# OpenReview forum: "Variation-Bounded Losses for Learning with Noisy Labels"
_ICLR.cc/2025/Conference — ICLR 2025 Conference Withdrawn Submission_

### Official Review · Reviewer_qKQr · 2024-10-29

**Soundness:** 3
**Presentation:** 3
**Contribution:** 2
**Rating:** 8
**Confidence:** 4

**Summary:**

This paper first proposed a novel metric called Variation Ratio to measure the robustness of loss functions based on the theoretical investigation that a smaller variation ratio would lead to better robustness. Then the authors proposed a family of robust loss functions called Variation-Bounded Losses for learning against label noise by leveraging this metric. Extensive experiments demonstrated that the proposed method outperforms existing robust loss functions over various benchmark datasets with different types of label noise.

**Strengths:**

1. This paper is well-written and easy to follow.
2. The proposed Variation-Bounded Losses are built on the property analysis on Variation Ratio, which is theoretically sound and practically relevant.
3. This paper conducted extensive experiments over various benchmark datasets and baseline approaches with different label noise and noise rates.

**Weaknesses:**

1. This paper introduced a list of hyperparameters (i.e., $\alpha$, $\beta$, and a) which varied across different datasets under different noise rates. It might limit the application scenarios.
2. The superiority of VBL compared with the other asymmetric loss functions is unclear.  As introduced in this paper, the common drawbacks of existing asymmetric loss functions are overly complicated with numerous hyperparameters and often suffer from underfitting. However, the proposed approach also needs a list of hyperparameters and does not have clear clarification about the underfitting issue.

**Questions:**

1. As the authors only provide hyperparameter details on CIFAR-10/100 in Table 6, how about the hyperparameter choice on datasets in Table 5? Could you please explain how to choose those hyperparameters across different datasets?

---

> ### Author Response · Authors · 2024-11-19
> **Rebuttal to Reviewer qKQr**
>
> Thank you very much for your positive comments. We would like to offer the following responses to your concerns.
>
> **Response to Weakness 1**
>
> Thank you for your valuable comment. We provide additional ablation experiments about NCE+VCE on CIFAR-100. The results over the last sota method are bolded.
>
> | CIFAR-100 | Clean | Symmetric 0.8 | Asymmetric 0.4 | Dependent 0.6 | Human |
> |:---:|:---:|:---:|:---:|:---:|:---:|
> | NCE+NNCE | 70.26±0.15 | 28.01±1.06 | 45.73±0.74 | 48.12±0.48 | 56.37±0.42 |
> | **($\alpha$=5, $\beta$=1, a=0.2)** | **72.06±0.70** | **33.61±0.91** | 45.35±0.43 | **51.09±1.01** | 55.94±0.23 |
> | **($\alpha$=5, $\beta$=1, a=0.4)** | **72.48±0.43** | **30.19±0.30** | **51.19±0.33** | **54.04±0.58** | **57.44±0.60** |
> | **($\alpha$=5, $\beta$=1, a=0.9)** | **71.70±0.11** | 15.51±0.11 | **55.82±0.99** | 46.20±0.43 | **58.75±0.32** |
>
> As can be seen, for fixed hyperparameters ($\alpha$=5, $\beta$=1, a=0.4), we can also significantly outperform the last sota method in all scenarios.
>
> **Response to Weakness 2**
>
> Thank you for your valuable comment.
>
> --- About hyperparameters
>
> Our VBL have a more concise form than the existing asymmetric loss functions.
> The existing asymmetric loss functions [1], AUL, AGCE, and AEL, all have two hyperparameters. Conversely, our VCE, VEL and VMSE have only one hyperparameter.
>
> --- About underfitting issue
>
> Our VBL have a more concise form, making them easier to adjust the hyperparameters and optimize.
> We provide more experimental demonstrations. "S" denotes symmetric noise, and "A" denotes asymmetric noise.  The top-3 best results are bolded.
>
> | CIFAR-10 | Clean | S (0.2) | S (0.4) | S (0.6) | S (0.8) | A (0.1) | A (0.2) | A (0.3) | A (0.4) |
> |:---:|:---:|:---:|:---:|:---:|:---:|:---:|:---:|:---:|:---:|
> | AUL | 91.27±0.12 | 89.21±0.09 | 85.64±0.19 | 78.86±0.66 | 52.92±1.20 | **90.19±0.16** | 88.17±0.11 | 84.87±0.04 | 56.33±0.07 |
> | AGCE | 88.95±0.22 | 86.98±0.12 | 83.39±0.17 | 76.49±0.53 | 44.42±0.74 | 88.08±0.06 | 86.67±0.14 | 83.59±0.15 | 60.91±0.20 |
> | AEL | 86.38±0.19 | 84.27±0.12 | 81.12±0.20 | 74.86±0.22 | 51.41±0.32 | 85.22±0.15 | 83.82±0.15 | 82.43±0.16 | 58.81±3.62 |
> | **VCE** | **91.63±0.09** | **90.31±0.23** | **87.62±0.19** | **82.22±0.34** | **64.25±0.41** | **90.69±0.23** | **89.46±0.14** | **85.29±0.15** | **75.48±3.85** |
> | **VEL** | **91.72±0.18** | **90.12±0.11** | **87.76±0.17** | **82.47±0.43** | **64.03±0.89** | **90.02±0.15** | **89.40±0.18** | **87.16±0.60** | **84.20±0.17** |
> | **VMSE** | **91.73±0.16** | **89.95±0.20** | **87.43±0.12** | **82.23±0.49** | **64.85±2.41** | 89.98±0.09 | **88.73±0.22** | **85.35±0.14** | **80.12±0.46** |
>
> As can be seen, our VBL significantly exceed the existing asymmetric losses, including the clean case. Since asymmetric losses and our VBL are theoretically proven to be noise-tolerant, the performance difference depends mainly on the fitting ability. The experimental results clearly demonstrate that our methods have a better fitting ability.
>
> We will clarify the advantages of our VBL over existing asymmetric loss functions more explicitly in the manuscript. Thank you again for your constructive feedback.
>
> **Response to Question 1**
>
>  Thank you for your kind comment.
>
> --- About hyperparameters for Table 5
>
> Please see Appendix B.2, Lines 797 and 805.
>
> --- About choosing hyperparameters
>
> For VBL, we can easily calculate a rough range based on Theorem 3, an example can be seen in Lines 242-244, then we will search hyperparameters based on empirical testing. For NCE+VBL, in most cases, we fix $\alpha=5, \beta=1$, and then search for $a$. In rare cases where $\alpha=5$ does not work very well, we also search for $\alpha$. Since $\beta$ is always fixed at 1, we actually have only two parameters to adjust. We will add these instructions to the manuscript for clarity.
>
> >[1] Asymmetric Loss Functions for Learning with Noisy Labels, ICML 2021.

---

> > ### Comment · Reviewer_qKQr · 2024-11-25
> >
> > Thank you for the author's response!
> >
> > My main concern about this paper is the hyperparameters. I think the authors have addressed my concerns. Thus, I raise my score to  8.

---

> > > ### Author Response · Authors · 2024-11-25
> > > **Thank you for your feedback**
> > >
> > > Dear Reviewer qKQr,
> > >
> > > Thank you very much for your feedback! We are pleased to address your concern. We truly appreciate your positive and insightful comments to help us improve the quality of our work.
> > >
> > > Best regards,
> > >
> > > The authors

---

> ### Author Response · Authors · 2024-11-24
> **Looking forward to your feedback**
>
> Dear Reviewer qKQr,
>
> We sincerely appreciate your valuable time and insightful comments on our work. As the deadline for the Author-Reviewer discussion draws near, we kindly request your feedback on whether our responses have effectively addressed your concerns. Thank you once again for your contributions, and we hope you have a wonderful day!
>
> Best regards,
>
> The Authors

---

### Official Review · Reviewer_DhVJ · 2024-10-30

**Soundness:** 4
**Presentation:** 3
**Contribution:** 3
**Rating:** 6
**Confidence:** 4

**Summary:**

This paper addresses the challenges of learning with noisy labels. The authors propose a new metric called the *Variation Ratio* to measure the robustness of loss functions. Based on this metric, a new family of robust loss functions, *Variation-Bounded Losses* (VBL), are proposed. The authors analyze the properties of variation-bounded losses theoretically and found that a smaller variation ratio can achieve better robustness. Compared to the symmetric condition, the variation ratio can provide a more relaxed condition to achieve noise tolerance. The experiments on several datasets (CIFAR-10, CIFAR-100, WebVision, ILSVRC12, and Clothing1M) demonstrate the effectiveness of the proposed method. Additionally, sensitivity tests for the hyperparameter $a$ are provided.

**Strengths:**

1. The authors proposed a new metric called variation ratio to measure the robustness of loss functions. This metric can provide a more relaxed condition to achieve noise tolerance.
2. Detailed properties of variation-bounded losses are analyzed theoretically.
3. The authors generalize commonly used loss functions (Cross Entropy, Exponential Loss and Mean Square Error) to the variation-bounded form.
4. Sufficient experiments are conducted. The authors not only evaluate the test accuracy of the proposed losses on the benchmark datasets but also provide sensitivity tests for the hyperparameter $a$.
5. The authors visualize the learned embeddings using t-SNE on the CIFAR-10 dataset. The visualization results show the embeddings for the proposed losses are well-separated clusters.

**Weaknesses:**

In the experiments conducted on the CIFAR-10 dataset, the authors report the results for VCE, VEL, and VMSE. However, the results of these methods for the CIFAR-100 and CIFAR-100N datasets are absent. Only the results of NCE-VCE, NCE-VEL, and NCE-VMSE are reported. This inconsistency in reporting across datasets may hinder a comprehensive evaluation of the methods' performance. It would be beneficial for the authors to explain the motivation behind this.

**Questions:**

In Figure 1, the test accuracy of VCE with $a=10$ is noted to be lower than that with $a=4$. The authors claim that a too small variation ratio may reduce the fitting ability. When the training epoch increases, whether the test accuracy of VCE $a=10$ can surpass that of VCE $a=4$?

---

> ### Author Response · Authors · 2024-11-19
> **Rebuttal to Reviewer DhVJ**
>
> Thank you very much for your positive comments. We would like to offer the following responses to your concerns.
>
> **Response to Weakness**
>
> Thank you for your valuable comment. In the following, we provide a detailed explanation about the combination of VBL with NCE.
>
> --- Why is the combination beneficial
>
> The combination of two different robust loss functions, such as VBL and NCE, can mutually enhance the optimization processes of each other, thus improving the overall fitting ability of the model. Previous works [1, 2, 3] have experimentally demonstrated that combining NCE often results in improved performance.
>
> --- Why use the combination for CIFAR-100
>
> In fact, NCE+VBL can achieve good performance on any dataset, not just CIFAR-100.
> To verify this, we conduct more experiments about NCE+VCE on other datasets, where "S" denotes symmetric noise and "A" denotes asymmetric noise.  The results over the last sota method are bolded.
>
> | CIFAR-10 | Clean | S (0.2) | S (0.4) | S (0.6) | S (0.8) | A (0.1) | A (0.2) | A (0.3) | A (0.4) |
> |:---:|:---:|:---:|:---:|:---:|:---:|:---:|:---:|:---:|:---:|
> | NCE+NNCE | 91.74±0.18 | 89.68±0.29 | 87.16±0.16 | 81.28±0.63 | 62.28±1.10 | 90.66±0.16 | 89.09±0.21 | 85.49±0.49 | 77.99±0.40 |
> | **VCE** | 91.63±0.09 | **90.31±0.23** | **87.62±0.19** | **82.22±0.34** | **64.25±0.41** | **90.69±0.23** | **89.46±0.14** | 85.29±0.15 | 75.48±3.85 |
> | **NCE+VCE** | 91.79±0.33 | **90.13±0.25** | **87.71±0.12** | **82.59±0.08** | **64.92±1.87** | **90.70±0.12** | **89.57±0.16** | **86.98±0.09** | **79.97±0.30** |
>
> | Loss 	| NCE+NNCE 	| VCE 	| NCE+VCE 	|
> |:---:|:---:|:---:|:---:|
> | WebVision | 67.44 | **70.12** | **69.36** |
> | ILSVRC12 	| 65.00 | **66.12** | **66.16**|
> | Clothing1M | 69.75 | **70.23** | **70.49**|
>
> As can be seen, NCE+VCE also significantly outperforms the last sota method on other datasets.
> Moreover,  VCE alone often surpasses the sota method in most scenarios other than CIFAR-100.
> Therefore, for datasets other than CIFAR-100, we opt to use VBL alone to better highlight the superiority of our method, even though it puts us at a disadvantage compared to methods that combine NCE.
>
> Since CIFAR-100 is a more challenging dataset, the improvement from incorporating NCE is more pronounced.
> If we do not combine with NCE, we cannot completely exceed the previous sota methods that also utilize NCE.
> Therefore, we opt to combine NCE with VBL on CIFAR-100 to surpass them.
>
> We will incorporate these more detailed interpretations into the manuscript to enhance clarity and completeness.
> Thank you once again for your valuable suggestions and for helping us improve our work.
>
> **Response to Question**
>
>  Thank you very much for your insightful comment. We provide experiments about more epochs on CIFAR-10 with 0.8 symmetric noise. The best results are bolded.
>
> | CIFAR-10 | 120 epochs | 200 epochs | 400 epochs |
> |:---:|:---:|:---:|:---:|
> | GCE (q=0.9) | 46.61±0.39 | 33.39±0.91  | 23.25±0.74 |
> | **VCE (a=4)** | **64.25±0.41** | **63.02±1.34** | 59.55±0.72 |
> | **VCE (a=10)** | 62.73±2.73 |  62.74±2.59  | **61.41±2.02** |
>
> The results demonstrate two points. (1) As $f_\eta$ converges to the global minimum of $R^\eta_{L}(f)$, GCE basically loses robustness. Conversely, VCE still retains excellent noise-tolerance. This is consistent with our theory. (2) Although VCE (a=10) did not exceed the best result of VCE (a=4, 120 epochs), VCE (a=10) achieved a better performance than VCE (a=4) in the case of both training 400 epochs. This shows that a smaller variation ratio provides better robustness, which is consistent with our theory.
>
> Thank you again for your insightful comment, which has enriched our work.
>
> >[1] Normalized Loss Functions for Deep Learning with Noisy Labels, ICML 2020.
> >
> >[2] Asymmetric Loss Functions for Learning with Noisy Labels, ICML 2021.
> >
> >[3] Active Negative Loss Functions for Learning with Noisy Labels, NeurIPS 2023.

---

> ### Author Response · Authors · 2024-11-24
> **Looking forward to your feedback**
>
> Dear Reviewer DhVJ,
>
> We sincerely appreciate your valuable time and insightful comments on our work. As the deadline for the Author-Reviewer discussion draws near, we kindly request your feedback on whether our responses have effectively addressed your concerns. Thank you once again for your contributions, and we hope you have a wonderful day!
>
> Best regards,
>
> The Authors

---

> > ### Comment · Reviewer_DhVJ · 2024-11-25
> >
> > Thank you for providing additional experimental results. These experimental results have significantly strengthened the paper. All my concerns are addressed.
> >
> > Best,
> >
> > Reviewer DhVJ

---

> > > ### Author Response · Authors · 2024-11-25
> > > **Thank you for your feedback**
> > >
> > > Dear Reviewer DhVJ,
> > >
> > > Thank you for your feedback! We are pleased to address your concerns and greatly appreciate your reviews, which play a crucial role in improving our work.
> > >
> > > Best regards,
> > >
> > > The authors

---

### Official Review · Reviewer_PV3A · 2024-11-02

**Soundness:** 2
**Presentation:** 2
**Contribution:** 2
**Rating:** 1
**Confidence:** 5

**Summary:**

In this article, authors focus on measuring the noise-tolerance of robust loss functions which are used to address noisy labels. The paper points out the limitations of existing robustness discriminative conditions, including symmetric and asymmetric conditions, and proposes a new metric (variation ratio) for loss functions. And authors theoretically analyze the relationship between the proposed metric and symmetric and asymmetric conditions. Additionally, this paper develops several robust losses based on the variation ratio.

**Strengths:**

This paper establishes a theoretical relationship between variation ratio and both symmetric and asymmetric conditions, demonstrating its effectiveness. Furthermore, variation ratio simplifies the discriminative conditions for asymmetric losses.

**Weaknesses:**

- The author analyzes the deficiencies of three kinds of robust losses in the introduction. However, the manuscript does not clearly indicate how the proposed loss functions address these issues, such as the slow convergence associated with symmetric loss functions.
- The author is suggested to emphasize the advantages of the proposed variation-based losses in comparison to loss functions satisfying symmetric or asymmetric conditions.
- The proposed variation ratio can not effectively measure the robustness of some typical robust losses, such as GCE mentioned in the manuscript, which exhibits robustness under noise rates exceeding 60%.

**Questions:**

- It appears that the proposed variation ratio is derived by dividing both the numerator and denominator of the asymmetric ratio [1] by $\Delta u$. However, the calculation of the variation ratio only relies on $L_{active}$, which may lose information about other classes.
- What’s the meaning of “partial noisy labels” in the introduction? It seems that the article lacks experiments about this noise type.

[1] 2021 AAAI Asymmetric Loss Functions for Learning with Noisy Labels

**Details Of Ethics Concerns:**

=========After Rebuttal=============
Overall, the following concerns have not well addressed in the Rebuttal.

- About Robustness

Variation ratio can’t measure the robustness of loss functions effectively. Specifically, the variation ratios of GCE and CE are ∞. However, theoretical and empirical studies have demonstrated that GCE is more robust than CE, which means a better metric should reflect the difference between above two losses. In an earlier response, the author stated that "The experiment demonstrated that, as epochs increase, GCE degrades to a performance level similar to CE". However, in a subsequent response, the author mentioned that "GCE does show better robustness than CE", which is inconsistent with the previous statement. Similarly, the variation ratio of JS loss proposed in [1] is also ∞.

- About global minimum

The theories of robust losses demonstrate the reason that applying robust losses can mitigate the negative impact of label noise by analyzing the global minimum of risk, which is instructive for the practical application of robust losses. However, using the final convergence state of the model to represent the global minimum is not appropriate for explaining practical problems.

- The relationship between asymmetric ratio

Although the proposed variation ratio v(l) can be applied to the loss function containing both active and passive term, the structure of variation ratio is similar to that of asymmetric ratio. In most situations, variation ratio $v(l)$ is the reciprocal of the asymmetric ratio r(l). For example, both the $v(l)$ and $r(l)$ of MAE are 1; the $v(l)$ of GCE is ∞ and the $r(l)$ of GCE is 0. Besides, we verify the above point on AEL in [2]. The $v(l) $ of AEL is $e^{(1/a)}$ and $r(l)$ of AEL is $e^{(-1/a)}$. As for AUL and AGCE in [2], when $q <1$, the $v(l) = 1/r(l)$; when $q>=1$, $v(l)$ can’t illustrate AUL and AGCE are completely asymmetric and the $r(l)$ of AUL and AGCE are 1.

[1] 2021 NIPS Generalized Jensen-Shannon Divergence Loss.

[2] 2021 ICML Asymmetric Loss Functions for Learning with Noisy Labels.

As a whole, I chose to reject this manuscript, as I feel confident that it fits. Overall, I believe above concerns must be well addressed to merit publication in a venue like this.

---

> ### Author Response · Authors · 2024-11-19
> **Rebuttal to Reviewer PV3A (1/2)**
>
> Thank you very much for your valuable comments. We would like to offer the following responses to your concerns.
>
> **Response to Weakness 1 and Weakness 2**
>
> --- Comparison to symmetric loss functions
>
> In the following, we explain why VBL address the slow convergence of symmetric loss functions
>
> 1) From the perspective of symmetric conditions, our method is actually a relaxation of the symmetric condition, which has a looser constraint than the symmetric losses. Our VBL, with bounded variation ratio, ensure sufficient and controlled gradient variation compared to symmetric losses like MAE. This enables our method to maintain robustness while significantly improving convergence speed.
>
> 2) From the perspective of the asymmetric condition, we clarify in Lines 241-248.
>
> 3) From the experimental perspective, we show symmetric losses in Table 1, such as MAE and NCE. As can be seen, our methods are significantly superior to these.
>
> --- Comparison to the hybrid between CE and MAE
>
> These hybrid loss functions, often result in reduced robustness due to their reliance on the CE component, as their variation ratio is $ \infty$. In contrast, our variation-bounded losses achieve complete noise tolerance while maintaining good fitting ability. We discuss these points in the paper, such as Lines 315–323.
>
> In addition, further response to your concern about the hybrid loss GCE can be found in the "Response to Weakness 3."
>
> --- Comparison to existing asymmetric loss functions:
>
> 1) Simplifying the complexity: Our VBL have a more concise form than the existing asymmetric loss functions.
> The existing asymmetric loss functions [1], AUL, AGCE, and AEL, all have two hyperparameters. Conversely, our VCE, VEL and VMSE have only one hyperparameter.
>
> 2) Better fitting ability: Our VBL have a more concise form, making them easier to adjust the hyperparameters and optimize.
> We provide more experimental demonstrations. "S" denotes symmetric noise, and "A" denotes asymmetric noise. The top-3 best results are bolded.
>
> | CIFAR-10 | Clean | S (0.2) | S (0.4) | S (0.6) | S (0.8) | A (0.1) | A (0.2) | A (0.3) | A (0.4) |
> |:---:|:---:|:---:|:---:|:---:|:---:|:---:|:---:|:---:|:---:|
> | AUL | 91.27±0.12 | 89.21±0.09 | 85.64±0.19 | 78.86±0.66 | 52.92±1.20 | **90.19±0.16** | 88.17±0.11 | 84.87±0.04 | 56.33±0.07 |
> | AGCE | 88.95±0.22 | 86.98±0.12 | 83.39±0.17 | 76.49±0.53 | 44.42±0.74 | 88.08±0.06 | 86.67±0.14 | 83.59±0.15 | 60.91±0.20 |
> | AEL | 86.38±0.19 | 84.27±0.12 | 81.12±0.20 | 74.86±0.22 | 51.41±0.32 | 85.22±0.15 | 83.82±0.15 | 82.43±0.16 | 58.81±3.62 |
> | **VCE** | **91.63±0.09** | **90.31±0.23** | **87.62±0.19** | **82.22±0.34** | **64.25±0.41** | **90.69±0.23** | **89.46±0.14** | **85.29±0.15** | **75.48±3.85** |
> | **VEL** | **91.72±0.18** | **90.12±0.11** | **87.76±0.17** | **82.47±0.43** | **64.03±0.89** | **90.02±0.15** | **89.40±0.18** | **87.16±0.60** | **84.20±0.17** |
> | **VMSE** | **91.73±0.16** | **89.95±0.20** | **87.43±0.12** | **82.23±0.49** | **64.85±2.41** | 89.98±0.09 | **88.73±0.22** | **85.35±0.14** | **80.12±0.46** |
>
> As can be seen, our VBL significantly exceed the existing asymmetric losses, including the clean case. Since asymmetric losses and our VBL are theoretically proven to be noise-tolerant, the performance difference depends mainly on the fitting ability. The experimental results clearly demonstrate that our methods have a better fitting ability.
>
> We will clarify the advantages of our VBL over other robust loss functions more explicitly in the manuscript. Thank you again for your constructive feedback.
>
> **Response to Weakness 3**
>
> Thank you for your valuable comment. Our theory measures the global minimization of the expected risk $R^\eta_{L}(f)$, i.e., the final convergence state of the model.
> Neural networks tend to fit clean labels at the early phases of training [2], hence the performance at the early phases of training cannot prove the noise-tolerance for the convergent state.
> We provide experiments about more epochs on CIFAR-10 with 0.8 symmetric noise. The best results are bolded.
>
> | CIFAR-10 | 120 epochs | 200 epochs | 400 epochs |
> |:---:|:---:|:---:|:---:|
> | GCE (q=0.9) | 46.61±0.39 | 33.39±0.91  | 23.25±0.74 |
> | **VCE (a=4)** | **64.25±0.41** | **63.02±1.34** | 59.55±0.72 |
> | **VCE (a=10)** | 62.73±2.73 | 62.74±2.59  | **61.41±2.02** |
>
> As shown, with the increase in epochs, as $f_\eta$ converges to the global minimum of $R^\eta_{L}(f)$, GCE basically loses robustness. Conversely, VCE still retains excellent noise-tolerance. This is consistent with our theory.

---

> > ### Author Response · Authors · 2024-11-19
> > **Rebuttal to Reviewer PV3A (2/2)**
> >
> > **Response to Question 1**
> >
> > Thank you for your comment.
> > Our variation ratio and the asymmetric ratio [1] are not equivalent.
> > The asymmetric ratio [1] only considers the loss functions that have only $\ell_{active}$ (see Definition 3 in [1]). Therefore, this does not lose information about other classes for the losses that have only $\ell_{active}$.
> >
> > In addition, our variation ratio can extend asymmetric losses to the loss that has both active and passive terms, i.e., VMSE. We prove theoretically that the asymmetric property of VMSE depends only on the variation ratio and not on $\ell_{passive}$ (see Corollary 1).
> >
> > **Response to Question 2**
> >
> > Thank you for your comment.
> > "partial noisy labels" means "a part of noisy labels", this is not a label noise type. We apologize for the ambiguity and have modified the ambiguous word in the manuscript.
> >
> > >[1] Asymmetric Loss Functions for Learning with Noisy Labels, ICML 2021.
> > >
> > >[2] Co-teaching: Robust Training of Deep Neural Networks with Extremely Noisy Labels, NeurIPS 2018.
> >
> > ---
> >
> > We appreciate your hard work during the review process and kindly ask you to reconsider the novelty and contributions of our work.

---

> > ### Comment · Reviewer_PV3A · 2024-11-26
> > **Official Comment by Reviewer PV3A**
> >
> > - Weakness 1:
> > Sufficient and controlled gradient variation can’t directly demonstrate the improvement of convergence speed.  The authors are suggested to illustrate the improvement of convergence speed from theoretical and experimental perspectives.
> >
> > - Weakness 3:
> > The variation ratios of GCE and CE are $\infty$. However, GCE loss demonstrates stronger robustness compared to CE, which means the proposed metric cannot measure the robustness of loss functions effectively. Besides，as epochs increases, $f_\eta$ may not converges to the global minimum.
> >
> > - For the proposed VCE in the paper, based on the parameter settings in Table 6, it can be observed that the values of loss functions are negative. This results in the loss functions losing their corresponding physical meanings.
> >
> > - The condition judging whether the loss function is asymmetric is $v(l)<w_t/w_i$, while in [1], the condition is $1/r(l)<w_t/w_i$. It seems that the variation ratio is the reciprocal of the asymmetric ratio $r(l)$. Could the authors provide further explanation of the relationship between the above two ratios from the formal perspective.
> >
> > [1] 2021 AAAI Asymmetric Loss Functions for Learning with Noisy Labels

---

> > > ### Author Response · Authors · 2024-11-26
> > > **Further Rebuttal to Reviewer PV3A (1/2)**
> > >
> > > Thank you for your feedback. We would like to offer the following responses to your new concerns.
> > >
> > > **Comment:** *Weakness 1: Sufficient and controlled gradient variation can’t directly demonstrate the improvement of convergence speed. The authors are suggested to illustrate the improvement of convergence speed from theoretical and experimental perspectives.*
> > >
> > > **Response:**
> > >
> > > --- From the theoretical perspective
> > >
> > > We conduct additional in-depth analysis from the perspective of optimization. Take VCE for example, the gradients of VCE and symmetric MAE are given by $-\frac{1}{u_y + a}$ and $-2$, respectively. This indicates that VCE is strictly convex, while MAE exhibits linearity. As a result, VCE demonstrates better optimization properties compared to symmetric MAE.
> > > As $a \to 0$ and the variation ratio increases, VCE exhibits stronger convexity. Specifically, the gradient of VCE changes more rapidly as $\frac{1}{(u_y + a)^2}$ increases.
> > >
> > > --- From the experimental perspective
> > >
> > > We provide additional ablation experiments about different variation ratios. In order to show the convergence speed more clearly, we used VCE to conduct experiments on the CIFAR-10 clean labels. The best results are bolded.
> > >
> > > | CIFAR-10 | 10 epochs | 20 epochs | 60 epochs | 120 epochs |
> > > |---|---|---|---|---|
> > > | VCE(a=0.2) | **81.47±0.27** | **85.73±0.52** | **90.21±0.08** | **91.69±0.10** |
> > > | VCE(a=4) | 76.06±1.19 | 80.63±1.06 | 86.17±0.67 | 91.63±0.09 |
> > > | VCE(a=10) | 75.47±0.68 | 78.45±1.86 | 83.66±0.64 | 90.88±0.25 |
> > >
> > > It is obvious that a sufficient variation ratio leads to better convergence speed.
> > >
> > > **Comment:** *Weakness 3: The variation ratios of GCE and CE are $\infty$. However, GCE loss demonstrates stronger robustness compared to CE, which means the proposed metric cannot measure the robustness of loss functions effectively. Besides, as epochs increases, may $f_\eta$ not converges to the global minimum.*
> > >
> > > **Response:**
> > >
> > > --- about the variation ratio of GCE
> > >
> > > We have included an additional experiment in our previous response. Please refer to **"Response to Weakness 3"** in the earlier response.
> > >
> > > The experiment demonstrated that, as epochs increases, GCE degrades to a performance level similar to CE (about 20\% accuracy). This result validates the effectiveness of our metric. Previous work [1] has also reached a similar conclusion regarding GCE.
> > >
> > > --- about $f_\eta$ may not converges to the global minimum.
> > >
> > > In deep learning, convergence to a global or local minimum is not critical, as the expected risks of all minimum are generally the same [2]. All the local minimum of the neural network are actually connected together, they actually belong to the same global minimum [3,4]. Moreover, much of the literature [1, 5] on other robust loss functions also assumes conditions under which the model converges to a global minimum.
> > >
> > > **Comment:** *For the proposed VCE in the paper, based on the parameter settings in Table 6, it can be observed that the values of loss functions are negative. This results in the loss functions losing their corresponding physical meanings.*
> > >
> > > **Response:**
> > > The meaning of the loss function is unrelated to whether it is negative or not, as long as it is decreasing. Many common loss functions are negative, such as unhinged loss [6] and negative cosine similarity in SimSiam [7]. If you want a positive loss function, you can add a positive constant, but it has no real impact in optimization.

---

> > > > ### Author Response · Authors · 2024-11-26
> > > > **Further Rebuttal to Reviewer PV3A (2/2)**
> > > >
> > > > **Comment:** *The condition judging whether the loss function is asymmetric is $v(l) < w_t/w_i$
> > > > , while in [1], the condition is $1/r(l) < w_t/w_i$
> > > > . It seems that the variation ratio is the reciprocal of the asymmetric ratio $r(l)$
> > > > . Could the authors provide further explanation of the relationship between the above two ratios from the formal perspective.*
> > > >
> > > > **Response:**
> > > > Our variation ratio $v(L)$ can be derived from the symmetric condition (specifically as the sum of losses for each class label, please see the proof of Lemma 1). Furthermore, we discovered that the variation ratio can also be used to derive the asymmetric condition through Lagrange's mean value theorem (please see the proof of Theorem 3).
> > > >
> > > > The asymmetry ratio $r(l)$ in [1] is essentially an equivalent formulation of the asymmetric condition (please see the proof of Theorem 6 in [1]). It is simply another way of expressing the asymmetric condition. Consequently, the asymmetry ratio has a complex form and lacks simplified the asymmetric condition, making it difficult to serve as a practical guide for designing simple and effective loss functions.
> > > >
> > > > Because our variation ratio $v(L)$ can derive the asymmetric condition, and the asymmetry ratio [1] is actually an equivalent representation of the asymmetric condition. So they look similar on the surface. In comparison, our concise variation ratio $v(L)$ is more suitable as a guide for designing robust loss functions. In addition, the asymmetry ratio can only be applied to loss functions containing only the active term, while our variation ratio $v(L)$ can be applied to the loss function containing both active and passive terms, i.e., VMSE.
> > > >
> > > > From the experimental perspective, abundant experiments have proved the effectiveness of our method compared to the asymmetry ratio.
> > > >
> > > > Additionally, we would like to point out a small typo in the reviewer's comment: the paper [1] was published in ICML, not AAAI.
> > > >
> > > > >[1] Asymmetric Loss Functions for Learning with Noisy Labels, ICML 2021.
> > > > >
> > > > >[2] Deep Learning without Poor Local Minima, NeurIPS 2016.
> > > > >
> > > > >[3] Essentially No Barriers in Neural Network Energy Landscape, ICML 2018.
> > > > >
> > > > >[4] On Connected Sublevel Sets in Deep Learning, ICML 2019.
> > > > >
> > > > >[5] Robust Loss Functions under Label Noise for Deep Neural Networks, AAAI 2017.
> > > > >
> > > > >[6] On the Dynamics Under the Unhinged Loss and Beyond, JMLR 2023.
> > > > >
> > > > >[7] Exploring Simple Siamese Representation Learning, CVPR 2021.
> > > >
> > > > ---
> > > > We hope that our response helps resolve your misunderstanding and concerns. We look forward to your feedback.

---

> > > > > ### Comment · Reviewer_PV3A · 2024-11-29
> > > > > **Official Comment by Reviewer PV3A-Second Round**
> > > > >
> > > > > - Although GCE is not an asymmetric loss, [1] theoretically exhibits a conclusion similar to Theorem 1, demonstrating that GCE is more robust than CE. In the following studies [2,3,4], GCE consistently outperforms CE across various datasets with different noise rates. However, it can not illustrate that GCE is more robust than CE by using variation ratio. It seems that variation ratio might serve as a better metric for determining whether a loss is asymmetric, but it may not effectively characterize robustness.
> > > > > -``The experiment demonstrated that, as epochs increases, GCE degrades to a performance level similar to CE (about 20% accuracy)" In your manuscript, GCE seems to behave more robust than CE. This result can also be observed in many papers. I do not think your point is reasonable.
> > > > >
> > > > >
> > > > > - Besides, it has been proved that for GCE, $R(f^*)-R(f^*_\eta)$ is bounded, where $f^*$ is the global minimum of $R(f)$ and $R(f)$ is calculated on the clean data. If we assume that the model converges to the global minimum as epochs increases, the test accuracy difference between $f$ and $f_\eta$ exceeds 60%, which is inconsistent with the theoretical result. It appears that the global minimum exists, but it is hard to acquire in practice.
> > > > >
> > > > > [1] 2018 NIPS Generalized Cross Entropy Loss for Training Deep Neural Networks with Noisy Labels.
> > > > > [2] 2020 ICML Normalized Loss Functions for Deep Learning with Noisy Labels.
> > > > > [3] 2021 NIPS Generalized Jensen-Shannon Divergence Loss.
> > > > > [4] 2023 NIPS Active Negative Loss Functions for Learning with Noisy Labels.

---

> > > > > > ### Author Response · Authors · 2024-11-29
> > > > > > **Response to your comments**
> > > > > >
> > > > > > Dear Reviewer PV3A
> > > > > >
> > > > > > Thank you for your prompt feedback. While we had previously conducted additional experiments on GCE and provided detailed explanations, it seems our message was not conveyed as clearly as intended. We have carefully addressed the reviewers' comments as outlined below:
> > > > > >
> > > > > > **Response to Comment 1:** In the following, we reply to the reviewer's misunderstanding about GCE results.
> > > > > > We re-paste the experiments in **"Response to Weakness 3"** below:
> > > > > >
> > > > > > | CIFAR-10 | 120 epochs | 200 epochs | 400 epochs |
> > > > > > |:---:|:---:|:---:|:---:|
> > > > > > | GCE (q=0.9) | 46.61±0.39 | 33.39±0.91  | 23.25±0.74 |
> > > > > > | **VCE (a=4)** | **64.25±0.41** | **63.02±1.34** | 59.55±0.72 |
> > > > > > | **VCE (a=10)** | 62.73±2.73 | 62.74±2.59  | **61.41±2.02** |
> > > > > >
> > > > > > **It can be seen that at 120 epochs (as used in our manuscript), GCE does show better robustness than CE.**
> > > > > > We do not dispute this and have explicitly demonstrated it in Figure 1 and the experimental results. One reason for this result is that, in the actual PyTorch implementation, probabilities cannot approach 0 infinitely closely (for example, they are truncated at 1e-7 [1,2,3] to prevent value overflows). If the minimum $u_y$ is 1e-7, the variation ratio of GCE (q=0.9) is not actually $\infty$ but approximately 5, whereas the variation ratio of CE is 1e7. This demonstrates that GCE is more robust.
> > > > > >
> > > > > > **As epochs increased, GCE degrades to a performance level about 20\% accuracy at 400 epochs (as used in our additional experiments in the response).** The variation ratio is 1.25 for VCE (a=4), and 1.1 for VCE (a=10). Therefore, VCE shows stronger robustness compared to GCE and CE.
> > > > > >
> > > > > > We acknowledge and respect the contributions of prior work [4] and have conducted careful comparisons in our manuscript.
> > > > > > Our study does not contradict their findings but instead provides a new theoretical perspective for reference.
> > > > > >
> > > > > >
> > > > > > **Response to Comment 2:** In the following, we reply to the reviewer's misunderstanding about the label noise model and the global minimum.
> > > > > >
> > > > > > --- about the label noise model
> > > > > >
> > > > > > The label noise model [1,2,3,4,5] assumes that labels randomly flip according to a specified noise probability. However, during the actual training phase, the labels remain fixed and do not flip randomly.
> > > > > > Even methods that are completely  noise-tolerant in theory will be affected by noisy labels in practical training.
> > > > > > Therefore, in practical applications, it is only possible to mitigate the impact of noisy labels rather than achieve the same accuracy as with clean labels.
> > > > > > For instance, advanced robust loss functions [1,2,3,4,5] typically achieve around 30\% accuracy on CIFAR-100 with 0.8 symmetric noise, compared to 70\% accuracy on clean labels.
> > > > > >
> > > > > > While we hope that the problem of noisy labels can be completely resolved in the future, no solution currently exists to fully address this issue.
> > > > > >
> > > > > > --- about the global minimum
> > > > > >
> > > > > > The explanation of the global minimum is provided solely in response to your question and does not impact the practical use.
> > > > > > We strictly followed the experimental setup of previous works [1,2,3], and our reproduced results were consistent with theirs. Extensive experiments have demonstrated the superiority of our method.
> > > > > >
> > > > > > >[1] Normalized Loss Functions for Deep Learning with Noisy Labels, ICML 2020.
> > > > > > >
> > > > > > >[2] Asymmetric Loss Functions for Learning with Noisy Labels, ICML 2021.
> > > > > > >
> > > > > > >[3] Active Negative Loss Functions for Learning with Noisy Labels, NeurIPS 2023.
> > > > > > >
> > > > > > >[4] Generalized Cross Entropy Loss for Training Deep Neural Networks with Noisy Labels, NeurIPS 2018.
> > > > > > >
> > > > > > >[5] Robust Loss Functions under Label Noise for Deep Neural Networks, AAAI 2017.
> > > > > >
> > > > > > ---
> > > > > >
> > > > > > We contend that rejecting our manuscript solely due to these misunderstandings does not fairly assess our contributions. Extensive experiments demonstrate the superiority of our method in mitigating various  label noise.
> > > > > > We hope that our response clarifies the reviewer's concerns, and we look forward to the reviewer's additional comments.
> > > > > >
> > > > > > Best regards,
> > > > > >
> > > > > > The Authors

---

> > > > > > > ### Author Response · Authors · 2024-12-01
> > > > > > > **Awaiting your further feedback**
> > > > > > >
> > > > > > > Dear Reviewer PV3A,
> > > > > > >
> > > > > > > The additional discussion period will come to an end. We kindly request your feedback on whether our new responses have effectively addressed your concerns.
> > > > > > >
> > > > > > > Best regards,
> > > > > > >
> > > > > > > The Authors

---

> ### Author Response · Authors · 2024-11-24
> **Looking forward to your feedback**
>
> Dear Reviewer PV3A,
>
> We sincerely appreciate your valuable time and insightful comments on our work. As the deadline for the Author-Reviewer discussion draws near, we kindly request your feedback on whether our responses have effectively addressed your concerns. Thank you once again for your contributions, and we hope you have a wonderful day!
>
> Best regards,
>
> The Authors

---

### Official Review · Reviewer_kQBp · 2024-11-02

**Soundness:** 2
**Presentation:** 2
**Contribution:** 2
**Rating:** 5
**Confidence:** 4

**Summary:**

This paper investigates new robust loss functions for label noise based on the concept of variation ratio. A smaller variation ratio can potentially enhance robustness, as evidenced by excess risk bounds. The best bounds are achieved when the symmetry condition is satisfied (corresponding to the smallest variation ratio). On the other side, larger ratios may lead to loss functions that are easier to train. Additionally, when the variation ratio is sufficiently small, it it shown that the asymmetry condition is met.

Experiments are conducted with three main loss functions—Variation Cross Entropy (VCE), Variation Exponential Loss (VEL), and Variation Mean Square Error (VMSE)—on standard benchmarks with symmetric, asymmetric, instance-dependent, and natural label noise.

**Strengths:**

1) The variation ratio is a novel idea and the paper introduces novel loss functions based on this idea.
2) The problem of fighting label noise is of great practical importance and developing robust loss functions that are easier to train is an interesting line of research.
3) Promising Empirical Performance: The empirical results are promising, with the proposed loss functions consistently outperforming previously introduced robust loss functions across multiple benchmarks.

**Weaknesses:**

1) Definition of the Variation Ratio: The variation ratio is defined in terms of a decomposition of a loss function as the sum of active and passive terms. However, this decomposition is not unique in general. For example, consider the Mean Absolute Error (MAE) loss, $1−u_y$​. It can be decomposed with an active term $l_{active}(u_y)=1−u_y$​ and passive terms $l_{passive}(u_k)=0$ for $k\neq y$. Alternatively, we could represent it as an active term $l_{active}(u_y)=0$ and passive terms $l_{passive}(u_k)=u_k$ for $k\neq y$, since $1−u_y=\sum_{k\neq y}u_k$​. In the first decomposition, the variation ratio is 1, while in the second decomposition, it becomes undefined (0/0). The variation ratio as currently defined in Definition 2 is not an intrinsic property of the loss function but rather depends on the chosen active-passive decomposition. This ambiguity suggests an issue in the definition of the variation ratio. This is very important to solve in my opinion since the variation ratio is the central concept of the paper.

2) Hyperparameter Tuning: In the previous work by [Ye,2023], hyperparameters for the loss functions were selected based on performance at 80% symmetric noise, and the same hyperparameters were applied across different noise rates and types (asymmetric, CIFAR-N). In contrast, the current paper re-tunes hyperparameters for each noise type, which could introduce a favorable bias in the comparisons with prior work. Additionally, the paper should specify the range of hyperparameters considered during tuning to get a sense of the cost for tuning the method. This is particularly relevant since the loss functions in this paper require three hyperparameters, compared to only two in [Ye,2023] and [Ma,2020] (excluding the L1-regularization parameter).

[Ye,2023]: Xichen Ye, Xiaoqiang Li, Songmin Dai, Tong Liu, Yan Sun, and Weiqin Tong. Active negative loss
functions for learning with noisy labels. In Thirty-seventh Conference on Neural Information
Processing Systems, 2023.

[Ma,2020]: Xingjun Ma, Hanxun Huang, Yisen Wang, Simone Romano, Sarah Erfani, and James Bailey. Normalized loss functions for deep learning with noisy labels. In International conference on machine
learning, pp. 6543–6553. PMLR, 2020.

**Questions:**

1) How can you solve the issue in the definition of the Variation Ratio described in the point 1) of the weaknesses?
2) Can you provide more clarification about the tuning of the hyperparameters for your loss functions?

---

> ### Author Response · Authors · 2024-11-19
> **Rebuttal to Reviewer kQBp**
>
> Thank you very much for your valuable comments. We would like to offer the following responses to your concerns.
>
> **Response to Weakness 1 and Question 1**
>
> Thank you very much for highlighting this ambiguity in our definition. In both Lemma 1 and Theorem 3, we have explicitly stated that we only consider cases where $\ell_{active}$ is monotonically decreasing. To clarify, we will also incorporate this condition into Definition 2. Thank you once again for your valuable input.
>
> **Response to Weakness 2 and Question 2**
>
> Thank you for your meaningful comment.
>
> --- About fixing the hyperparameters across different noise scenarios
>
> We appreciate your valuable suggestion. As suggested, we provide additional experiments about NCE+VCE with fixed same hyperparameters across different noise scenarios. The results over the last sota method are bolded.
>
> | CIFAR-100 | Clean | Symmetric 0.8 | Asymmetric 0.4 | Dependent 0.6 | Human |
> |:---:|:---:|:---:|:---:|:---:|:---:|
> | NCE+NNCE | 70.26±0.15 | 28.01±1.06 | 45.73±0.74 | 48.12±0.48 | 56.37±0.42 |
> | **($\alpha$=5, $\beta$=1, a=0.2)** | **72.06±0.70** | **33.61±0.91** | 45.35±0.43 | **51.09±1.01** | 55.94±0.23 |
> | **($\alpha$=5, $\beta$=1, a=0.4)** | **72.48±0.43** | **30.19±0.30** | **51.19±0.33** | **54.04±0.58** | **57.44±0.60** |
> | **($\alpha$=5, $\beta$=1, a=0.9)** | **71.70±0.11** | 15.51±0.11 | **55.82±0.99** | 46.20±0.43 | **58.75±0.32** |
>
> As can be seen, for fixed hyperparameters ($\alpha=5$, $\beta=1$, $a=0.4$), we can also significantly outperform the last sota method in all scenarios.
>
> --- About the range of hyperparameters selection
>
> For NCE+VBL, in most cases, we fix $\alpha=5, \beta=1$, and then search for $a$. In rare cases where $\alpha=5$ does not work very well, we also search for $\alpha$. Since $\beta$ is always fixed at 1, we actually have only two parameters to adjust, consistent with previous works [1, 2]. We will add these instructions to the manuscript for clarity.
>
> >[1] Normalized Loss Functions for Deep Learning with Noisy Labels, ICML 2020.
> >
> >[2] Active Negative Loss Functions for Learning with Noisy Labels, NeurIPS 2023.

---

> ### Author Response · Authors · 2024-11-24
> **Looking forward to your feedback**
>
> Dear Reviewer kQBp,
>
> We sincerely appreciate your valuable time and insightful comments on our work. As the deadline for the Author-Reviewer discussion draws near, we kindly request your feedback on whether our responses have effectively addressed your concerns. Thank you once again for your contributions, and we hope you have a wonderful day!
>
> Best regards,
>
> The Authors

---

> > ### Author Response · Authors · 2024-12-01
> > **Awaiting your feedback**
> >
> > Dear Reviewer kQBp,
> >
> > The additional discussion period is also coming to an end. However, we have not yet received your feedback. We are eager to hear any additional comments or questions you may have regarding our work, as your insights have been invaluable to us.
> >
> > Best regards,
> >
> > The Authors

---

> ### Comment · Reviewer_kQBp · 2024-12-02
> **Thank you for your response**
>
> I thank you for your efforts in answering my questions. I still have the following concerns:
>
> 1.It seems that adding the assumption that $l_{active}$  is monotonically decreasing does not solve the problem in Definition 2. The problem comes from the fact that the active/passive decomposition is not unique. Consider the exponential loss $L=e^{-u_y}$. If we choose $l_{active}(u_y)=e^{-u_y}-u_y$ and $l_{passive}(u_k)=\frac{1}{K-1}-u_k$, we get $v(L)=\frac{max|-e^{-u_y}-1|}{min|-e^{-u_y}-1|}=\frac{max(e^{-u_y}+1)}{min(e^{-u_y}+1)}= \frac{2}{1+e^{-1}}$. However, this is different from $v(L)=e$ obtained from $l_{active}(u_y)=e^{-u_y}$ and $l_{passive}(u_k)=0$.
>
> Considering only loss functions with $l_{passive}(u_k)$ set to $0$ would solve this problem (like in Lemma 1 and Theorem 3). However, the decomposition active/passive would then not be relevant to the method of the paper. If this solution is used, some modifications would be required in the text. For example,  starting in line 296: ''To date, no studies have explored asymmetric loss
> functions that have both active and passive terms, because this is a more complex scenario. In this
> paper, we extend the asymmetric loss function to include both active and passive terms''.
>
> 2.Thank you for providing results showing that good performance can still be achieved on CIFAR100 without too much tuning. It is claimed in your response that ''Since $\beta$ is always fixed at 1, we actually have only two parameters to adjust''. This is true on the dataset CIFAR100. However, the parameter $\beta$ is different on CIFAR10 from what I can see in Table 6. This means that the parameter $\beta$ is actually being tuned. In my opinion, two reasonable solutions could be:
>
> a) Fix $\beta$ to 1 for all experiments and then the claim ''we actually have only two parameters to adjust, consistent with previous works'' is correct.
>
> b) Accepting that the method has more hyperparameters to tune and argue that the search of hyperparamters is still fair by, for example, comparing the size of the grid search among the different methods.

---

> ### Author Response · Authors · 2024-12-03
> **Further Rebuttal to Reviewer kQBp**
>
> Dear Reviewer kQBp,
>
> Thank you for your feedback. We would like to offer the following responses to your concerns.
>
> **Response to Comment 1:**
> Thank you for your kind comment. In fact, there is no ambiguity here because, in our definition, $L$ belongs to $\mathcal{L}$, which is defined in Lines 99-101. The definition of $\mathcal{L}$ shows that the loss function $L$ is formulated as $L(\mathrm {u},\mathrm{e_y})=\ell(u_y,1)+\sum_{k\neq y} \ell(u_k,0)$. Therefore, the terms $\ell_{active}(u_y)$ and $\ell_{passive}(u_k)$ share the same basis loss function, ensuring the decomposition is unique.
>
> **Response to Comment 2:**
> Thank you for your thoughtful suggestion. For CIFAR-10, the parameter $\beta$ denotes a scalar parameter, and we adjust this scalar parameter and $a$. Therefore, for CIFAR-10, we also only need to tune two parameters.
>
> For your solution (1), we will include experimental results with the fixed scalar parameter of 1 for VBL alone in the final version.
>
> For your solution (2), we acknowledge that, regardless of the specific experimental setting, our NCE+VBL involve three hyperparameters. This is more concise than the previous work [1], but they introduce one additional hyperparameter compared to previous works [2, 3].
>
> >[1] Asymmetric Loss Functions for Learning with Noisy Labels, ICML 2021.
> >
> >[2] Normalized Loss Functions for Deep Learning with Noisy Labels, ICML 2020
> >
> >[3] Active Negative Loss Functions for Learning with Noisy Labels, NeurIPS 2023.
>
> We hope this message finds you well. The discussion period is coming to an end and we look forward to your feedback.
>
> Best regards,
>
> The Authors

---

### Official Review · Reviewer_MvQX · 2024-11-03

**Soundness:** 3
**Presentation:** 3
**Contribution:** 3
**Rating:** 8
**Confidence:** 4

**Summary:**

The manuscript introduces the variation ratio, a new metric designed to control label noise in supervised learning. Label noise – which results from human error or incomplete labeling – often degrades the performance of deep neural networks. By using the variation ratio to evaluate the robustness of loss functions, the authors develop a new family of robust loss functions called variation-bounded losses (VBL). These functions have bounded variation ratios, and the work provides theoretical proof that a lower variational ratio results in higher noise tolerance. This approach provides a more flexible alternative to conventional symmetric loss functions, which often have a reduced fitting ability.

The authors generalize commonly used loss functions to a  variation bounded form, such as Variation Cross Entropy (VCE), Variation Exponential Loss (VEL), and Variation Mean Square Error (VMSE). These adaptations aim to retain the effectiveness and simplicity of the original loss functions while improving their robustness to label noise. Theoretical analyses confirm that VBLs achieve robustness without the added complexity of many hyperparameters typically found in asymmetric loss functions.

Extensive experiments demonstrate the practical advantages of variation-bounded losses in various datasets with synthetic and real-world types of noise. The results show that VBLs in many cases outperform other robust loss functions, achieving higher accuracy and robustness to noise. Furthermore, VBLs excel in scenarios involving real-world noisy datasets, underscoring their applicability beyond synthetic benchmarks. The study concludes that variation bounded losses not only improve the noise tolerance of models, but also provide a structured, less complex way to design effective loss functions in real-world applications with noisy labels.

**Strengths:**

The paper introduces a the concept of variation bounded losses to asses the noise tolerance of loss functions.

The paper establishes a connection between variation boundedness and both symmetric and asymmetric conditions.

New variation bounded variants of well known loss functions are formulated.

Based on a series of numerical experiments the authors demonstrate that the newly suggested loss functions improve on the state of the art in learning data sets with label noise.

**Weaknesses:**

I do not understand how the hyperparameters for combining NCE with the different VBLs are determined.

The paper would benefit from an explanation why the combination of VBLs with NCE is beneficial - this is important since this combination is most successful with CIFAR100. Is the combination still variation bounded?

**Questions:**

Can the authors  provide details on their hyperparameter selection process for combining NCE with VBLs? For example, did they use a specific optimization method like grid search or random search, or was it based on empirical testing? This information would help readers reproduce the results and understand the practical implementation

Can the authors provide a theoretical analysis or intuitive explanation for why combining VBLs with NCE is particularly effective, especially for CIFAR100? Additionally,  clarification on whether this combination preserves the variation-bounded property would be valuable for understanding the theoretical underpinnings of their approach.

---

> ### Author Response · Authors · 2024-11-19
> **Rebuttal to Reviewer MvQX**
>
> Thank you very much for your positive comments. We would like to offer the following responses to your concerns.
>
> **Response to Weakness 1 and Question 1**
>
> Thank you for your kind comment. We determine hyperparameters $\alpha$, $\beta$, $a$ for NCE+VBL based on empirical testing. Specifically, in most cases, we fix $\alpha=5, \beta=1$, then search for $a$. In rare cases where $\alpha=5$ does not work very well, we also search for $\alpha$. Since $\beta$ is always fixed at 1, there are actually only two parameters to adjust. We will add these instructions to the manuscript for clarity.
>
> **Response to Weakness 2 and Question 2**
>
> Thank you for your valuable comment.
>
> --- Why is the combination beneficial
>
> The combination of two different robust loss functions, such as VBL and NCE, can mutually enhance the optimization processes of each other, thus improving the overall fitting ability of the model. Previous works [1, 2, 3] have experimentally demonstrated that combining NCE often results in improved performance.
>
> --- Why use the combination for CIFAR-100
>
> In fact, NCE+VBL can achieve good performance on any dataset, not just CIFAR-100.
> To verify this, we conduct more experiments about NCE+VCE on other datasets, where "S" denotes symmetric noise and "A" denotes asymmetric noise.  The results over the last sota method are bolded.
>
> | CIFAR-10 | Clean | S (0.2) | S (0.4) | S (0.6) | S (0.8) | A (0.1) | A (0.2) | A (0.3) | A (0.4) |
> |:---:|:---:|:---:|:---:|:---:|:---:|:---:|:---:|:---:|:---:|
> | NCE+NNCE | 91.74±0.18 | 89.68±0.29 | 87.16±0.16 | 81.28±0.63 | 62.28±1.10 | 90.66±0.16 | 89.09±0.21 | 85.49±0.49 | 77.99±0.40 |
> | **VCE** | 91.63±0.09 | **90.31±0.23** | **87.62±0.19** | **82.22±0.34** | **64.25±0.41** | **90.69±0.23** | **89.46±0.14** | 85.29±0.15 | 75.48±3.85 |
> | **NCE+VCE** | 91.79±0.33 | **90.13±0.25** | **87.71±0.12** | **82.59±0.08** | **64.92±1.87** | **90.70±0.12** | **89.57±0.16** | **86.98±0.09** | **79.97±0.30** |
>
>
> | Loss 	| NCE+NNCE 	| VCE 	| NCE+VCE 	|
> |:---:|:---:|:---:|:---:|
> | WebVision | 67.44 | **70.12** | **69.36** |
> | ILSVRC12 	| 65.00 | **66.12** | **66.16**|
> | Clothing1M | 69.75 | **70.23** | **70.49**|
>
> As can be seen, NCE+VCE also significantly outperforms the last sota method on other datasets.
> Moreover,  VCE alone often surpasses the sota method in most scenarios other than CIFAR-100.
> Therefore, for datasets other than CIFAR-100, we opt to use VBL alone to better highlight the superiority of our method, even though it puts us at a disadvantage compared to methods that combine NCE.
>
> Since CIFAR-100 is a more challenging dataset, the improvement from incorporating NCE is more pronounced.
> If we do not combine with NCE, we cannot completely exceed the previous sota methods that also utilize NCE.
> Therefore, we opt to combine NCE with VBL on CIFAR-100 to surpass them.
>
> --- Theoretical guarantee for NCE+VBL
>
> According to our definition, variation ratio is suitable for loss functions where $\ell_{active}$ only depends on $u_y$.
> Since the $\ell_{active}$ of NCE depends not only on $u_y$ but also on $u_{k\neq y}$, NCE is not a function considered by the variation ratio. However, we can still prove that NCE+VBL is noise-tolerant.
>
> Previous work [2] proved that symmetric loss functions are completely asymmetric, and the combination of asymmetric loss functions remains asymmetric. Because NCE is symmetric (i.e., also asymmetric), and we have already proved that VBL is asymmetric. So NCE+VBL is still asymmetric and therefore noise-tolerant.
>
> We will incorporate these more detailed interpretations into the manuscript to enhance clarity and completeness.
> Thank you once again for your valuable suggestions and for helping us improve our work.
>
>
> >[1] Normalized Loss Functions for Deep Learning with Noisy Labels, ICML 2020.
> >
> >[2] Asymmetric Loss Functions for Learning with Noisy Labels, ICML 2021.
> >
> >[3] Active Negative Loss Functions for Learning with Noisy Labels, NeurIPS 2023.

---

> > ### Author Response · Authors · 2024-11-24
> > **Looking forward to your feedback**
> >
> > Dear Reviewer MvQX,
> >
> > We sincerely appreciate your valuable time and insightful comments on our work. As the deadline for the Author-Reviewer discussion draws near, we kindly request your feedback on whether our responses have effectively addressed your concerns. Thank you once again for your contributions, and we hope you have a wonderful day!
> >
> > Best regards,
> >
> > The Authors

---

> > ### Comment · Reviewer_MvQX · 2024-11-25
> >
> > Thank you for the clarifications, I maintain my score. Please incorporate the clarifications into a revised version of the manuscript.

---

> > > ### Author Response · Authors · 2024-11-25
> > > **Thank you for your feedback**
> > >
> > > Dear Reviewer MvQX,
> > >
> > > Thank you for approving our responses! We will incorporate these clarifications  into the final version. Your guidance has been instrumental in enhancing the quality of our work.
> > >
> > > Best regards,
> > >
> > > The authors

---

> > > > ### Comment · Reviewer_MvQX · 2024-11-26
> > > >
> > > > Dear Authors,
> > > >
> > > > can you please add the additional results and discussion in response to my (and possibly other) questions to the manuscript before the end of the discussion period?
> > > >
> > > > Best regards, Reviewer MvQX

---

> > > > > ### Author Response · Authors · 2024-11-26
> > > > > **Thank you for your feedback**
> > > > >
> > > > > Dear Reviewer MvQX,
> > > > >
> > > > > Thank you for your feedback! We will update the PDF within two days, by the end of November 27th (AoE).
> > > > >
> > > > > Best regards,
> > > > >
> > > > > The authors

---

> > > > > > ### Author Response · Authors · 2024-11-27
> > > > > > **We have submitted a revised PDF**
> > > > > >
> > > > > > Dear Reviewer MvQX,
> > > > > >
> > > > > > We have submitted a revised PDF, incorporating responses into different sections, mainly including the methodology section, the experiment section, and the appendix.
> > > > > >
> > > > > > In addition, we update some experiments on CIFAR-100.
> > > > > > Now, for all CIFAR-100 scenarios, we fix $\alpha = 5$ and $\beta = 1$. There is no need to adjust $\alpha$, and it consistently maintains good performance. This highlights the parameter tolerance and ease of use of NCE+VBL.
> > > > > >
> > > > > > Best regards,
> > > > > >
> > > > > > The authors

---

### Note · Authors · 2025-01-24

I have read and agree with the venue's withdrawal policy on behalf of myself and my co-authors.